# Change Point Detection in Dynamic Graphs with Decoder-only Latent Space Model

**Yik Lun Kei**                                          *ykei@ucsc.edu*
*Department of Statistics*
*University of California, Santa Cruz*

**Jialiang Li**                                          *jl2356@njit.edu*
*Department of Computer Science*
*New Jersey Institute of Technology*

**Hangjian Li**                                  *hangjian.li@walmart.com*
*Walmart Global Tech*

**Yanzhen Chen**                                     *imyanzhen@ust.hk*
*Department of Information Systems, Business Statistics and Operations Management*
*Hong Kong University of Science and Technology*

**Oscar Hernan Madrid Padilla**                *oscar.madrid@stat.ucla.edu*
*Department of Statistics and Data Science*
*University of California, Los Angeles*

**Reviewed on OpenReview:** *https://openreview.net/forum?id=DVeFqV56Iz*

## Abstract

This manuscript studies the unsupervised change point detection problem in time series of graphs using a decoder-only latent space model. The proposed framework consists of learnable prior distributions for low-dimensional graph representations and of a decoder that bridges the observed graphs and latent representations. The prior distributions of the latent spaces are learned from the observed data as empirical Bayes to assist change point detection. Specifically, the model parameters are estimated via maximum approximate likelihood, with a Group Fused Lasso regularization imposed on the prior parameters. The augmented Lagrangian is solved via Alternating Direction Method of Multipliers, and Langevin Dynamics are recruited for posterior inference. Simulation studies show good performance of the latent space model in supporting change point detection and real data experiments yield change points that align with significant events.

## 1 Introduction

Networks are often used to represent relational phenomena in numerous domains (Dwivedi et al., 2021; He et al., 2023; Han et al., 2023) and relational phenomena by nature progress in time. In recent decades, a plethora of dynamic network models has been proposed to analyze the interaction between entities over time, including Temporal Exponential Random Graph Model (Hanneke et al., 2010; Krivitsky & Handcock, 2014), Stochastic Actor-Oriented Model (Snijders, 2001; Snijders et al., 2010), and Relational Event Model (Butts, 2008; Butts et al., 2023). Although these interpretable models incorporate the temporal aspect for network analysis, network evolution is usually time-heterogeneous. Without taking the structural changes across dynamic networks into account, learning from the time series may lead to ambiguity, by confounding the structural patterns before and after a change happens. Hence, it is practical for social scientists to first

localize the change points in time series, and then study the networks within intervals, where no substantial change dilutes the network effects of interest.

More recently, considerable attention has been directed toward methodologies for change point detection in dynamic networks. Chen et al. (2020a) and Shen et al. (2023) employed embedding methods to detect both anomalous graphs and vertices in time series of networks. Park & Sohn (2020) combined the multi-linear tensor regression model with a hidden Markov model, detecting changes based on the transition between the hidden states. Sulem et al. (2023) learned a graph similarity function using a Siamese graph neural network to differentiate the graphs before and after a change point. Zhao et al. (2019) developed a screening algorithm that is based on an initial graphon estimation to detect change points. Huang et al. (2020) utilized the singular values of the Laplacian matrices as graph embedding to detect the differences across time. Chen & Zhang (2015), Chu & Chen (2019), and Song & Chen (2022a) proposed a non-parametric approach to delineate the distributional differences over time. Garreau & Arlot (2018) and Song & Chen (2022b) exploited the patterns in high dimensions via a kernel-based method. Madrid Padilla et al. (2022) identified change points by estimating the latent positions of Random Dot Product Graph (RDPG) models and by using a non-parametric version of the CUSUM statistic. Zhang et al. (2024) jointly trained a Variational Graph Auto-Encoder and a Gaussian Mixture Model to detect change points. Chen et al. (2024) and Athreya et al. (2024) considered network evolution in the Euclidean space and showed that the associated spectral estimates can localize the change points in network time series.

Inherently, network structures can be complex due to highly dyadic dependency. Acquiring a low-dimensional representation of a graph can summarize the enormous individual relations to promote downstream analysis (Hoff et al., 2002; Handcock et al., 2007; Gallagher et al., 2021). In particular, Larroca et al. (2021), Marenco et al. (2022), Zhu et al. (2023), Chen et al. (2024), and Athreya et al. (2025) studied different latent space models for dynamic graphs and focused on using node-level representation to detect changes in dynamic graphs. Furthermore, Sharifnia & Saghaei (2022) and Kei et al. (2023b) proposed to detect changes using an Exponential Random Graph Model (ERGM), which relies on user-specified network statistics to describe the structural patterns a priori. Given the complexity of dynamic network patterns, detecting change points in the data space can be challenging. Extending the framework of representation learning and network statistics, we aim to infer the graph-level representations that induce the structural changes in the latent space to support change point detection.

In addition, generative frameworks recently showed promising results in myriad applications, such as text generation with Large Language Model (Devlin et al., 2018; Lewis et al., 2019) and image generation with Diffusion Model (Ho et al., 2020; Rombach et al., 2022). Different from a graph generation task or network modeling task, we aim to explore how generative frameworks can assist the change point detection task for dynamic graphs. Specifically, Simonovsky & Komodakis (2018) proposed a Graph Variational Auto-Encoder (VAE) for graph generation, with a zero-mean Gaussian prior to regularize the latent space of graph-level representations. In the VAE framework (Kingma, 2013; Kipf & Welling, 2016; Lee et al., 2017; Bhattacharyya et al., 2018), the regularization via Kullback Leibler (KL) divergence arises from the Evidence Lower Bound (ELBO) for the marginal likelihood, encouraging the approximate posterior to be close to the fixed zero-mean Gaussian prior. Different from the VAE framework which involves an encoder, we focus on learning the mean of the Gaussian prior, with a decoder-only latent space model. In particular, we impose a Group Fused Lasso (GFL) regularization to the sequential differences of the prior parameters, so that the priors learned by minimizing the multivariate total variation can facilitate change point detection.

To exploit representation learning and generative frameworks for change point detection in dynamic graphs, we make the following contributions in this manuscript:

- We develop a decoder-only architecture to bridge the observed networks and latent variables for our change point detection method. We assume the graphs are generated from the latent variables that follow Gaussian prior distributions. With the graph decoder, the latent variables are considered as the graph-level representations of the observed networks.

- The parameters of the Gaussian priors for graph-level representations are learned to facilitate change point detection. Specifically, we apply Group Fused Lasso (GFL) regularization to promote sparsity

in the sequential differences of the multivariate prior parameters, effectively smoothing out minor fluctuations and highlighting significant change points.

- We derive an Alternating Direction Method of Multipliers (ADMM) procedure to solve the optimization problem associated with our method. Without an encoder, the model parameters are learned by inferring from the posterior via Langevin Dynamics. Experiments show good performance of the latent space model in supporting change point detection.

The rest of the manuscript is organized as follows. Section 2 specifies the proposed framework. Section 3 presents the objective function with Group Fused Lasso regularization and the ADMM procedure to solve the optimization problem. Section 4 discusses change points localization and model selection. Section 5 illustrates the proposed method on simulated and real data. Section 6 concludes the work with a discussion on the limitation and potential future developments.

## 2 Latent Space Model for Change Point Detection

### 2.1 Model Specification

For a node set $N = \{1, 2, \cdots, n\}$, we use an adjacency matrix $\boldsymbol{y} \in \{0, 1\}^{n \times n}$ to represent a graph or network. We denote the set of all possible node pairs as $\mathbb{Y} = N \times N$. In the adjacency matrix, $\boldsymbol{y}_{ij} = 1$ indicates an edge between nodes $i$ and $j$, while $\boldsymbol{y}_{ij} = 0$ indicates no edge. The relations can be either undirected or directed. The undirected variant has $\boldsymbol{y}_{ij} = \boldsymbol{y}_{ji}$ for all $(i, j) \in \mathbb{Y}$. Denote $\boldsymbol{y}^t$ as a network at a discrete time point $t$. The observed data is a sequence of networks $\boldsymbol{y}^1, \ldots, \boldsymbol{y}^T$.

For each network $\boldsymbol{y}^t \in \{0, 1\}^{n \times n}$, we assume there is a latent variable $\boldsymbol{z}^t \in \mathbb{R}^d$ such that the network $\boldsymbol{y}^t$ is generated from the latent variable with the following graph decoder:

$$\boldsymbol{y}^t \sim P(\boldsymbol{y}^t | \boldsymbol{z}^t) = \prod_{(i,j) \in \mathbb{Y}} \text{Bernoulli}(\boldsymbol{y}_{ij}^t; \boldsymbol{r}_{ij}(\boldsymbol{z}^t))$$

where $\boldsymbol{r}_{ij}(\boldsymbol{z}^t) = P(\boldsymbol{y}_{ij}^t = 1 | \boldsymbol{z}^t)$ is the Bernoulli parameter for dyad $\boldsymbol{y}_{ij}^t$ and it is elaborated in Section 2.3. Conditioning on the latent variable $\boldsymbol{z}^t$, we assume the network $\boldsymbol{y}^t$ is dyadic independent. We also impose a learnable Gaussian prior to the latent variable as

$$\boldsymbol{z}^t \sim P(\boldsymbol{z}^t) = \mathcal{N}(\boldsymbol{z}^t; \boldsymbol{\mu}^t, \boldsymbol{I}_d)$$

where $\boldsymbol{\mu}^t \in \mathbb{R}^d$ is the mean vector to be learned and $\boldsymbol{I}_d$ is an identity matrix. With graph decoder $P(\boldsymbol{y}^t | \boldsymbol{z}^t)$, we consider $\boldsymbol{z}^t \in \mathbb{R}^d$ as a graph-level representation for $\boldsymbol{y}^t \in \{0, 1\}^{n \times n}$. In this work, we estimate the prior parameters $\{\boldsymbol{\mu}^t\}_{t=1}^T$ to facilitate change point detection in $\{\boldsymbol{y}^t\}_{t=1}^T$.

### 2.2 Change Points

Anchored on the proposed framework, we can specify the change points to be detected, in terms of the prior parameters $\boldsymbol{\mu}^t \in \mathbb{R}^d$ for $t = 1, \ldots, T$. Let $\{C_k\}_{k=0}^{K+1} \subset \{1, 2, \ldots, T\}$ be a collection of ordered change points with $1 = C_0 < C_1 < \cdots < C_K < C_{K+1} = T$ such that

$$\boldsymbol{\mu}^{C_k} = \boldsymbol{\mu}^{C_k+1} = \cdots = \boldsymbol{\mu}^{C_{k+1}-1}, \ \ k = 0, \ldots, K,$$

$$\boldsymbol{\mu}^{C_k} \neq \boldsymbol{\mu}^{C_{k+1}}, \ \ k = 0, \ldots, K-1, \ \text{and} \ \boldsymbol{\mu}^{C_{K+1}} = \boldsymbol{\mu}^{C_K}.$$

The associated multiple change point detection problem comprises recovering the collection $\{C_k\}_{k=1}^K$ from a sequence of observed networks $\{\boldsymbol{y}^t\}_{t=1}^T$, where the number of change points $K$ is also unknown. In practice, change point detection problem is often discussed in an unsupervised manner.

In this work, to facilitate change point detection for $\{\boldsymbol{y}^t\}_{t=1}^T$ in the data space, we turn to learn the prior parameters $\{\boldsymbol{\mu}^t\}_{t=1}^T$ in the latent space. Intuitively, the consecutive prior parameters $\boldsymbol{\mu}^t$ and $\boldsymbol{\mu}^{t+1}$ are similar when no change occurs, but they are different when a change emerges. For notational simplicity, we denote $\boldsymbol{\mu} \in \mathbb{R}^{T \times d}$ as a matrix where the $t$-th row corresponds to $\boldsymbol{\mu}^t \in \mathbb{R}^d$ with $t = 1, \ldots, T$.

## 2.3 Choice of Graph Decoder

To facilitate change point detection in dynamic graphs, we choose to use the graph decoder that is standard and common in the literature (Kipf & Welling, 2016; Hamilton et al., 2017; Pan et al., 2018; Yang et al., 2019; Chen et al., 2020b; Wang et al., 2021b). Specifically, the graph decoder $P(\boldsymbol{y}^t|\boldsymbol{z}^t)$ is formulated with a Bernoulli parameter for dyad $\boldsymbol{y}_{ij}^t \in \{0,1\}$ as

$$\boldsymbol{r}_{ij}(\boldsymbol{z}^t) = P(\boldsymbol{y}_{ij}^t = 1|\boldsymbol{z}^t) = \boldsymbol{g}_{ij}\big(\boldsymbol{h}(\boldsymbol{z}^t)\big) \; \forall \; (i,j) \in \mathbb{Y}.$$

The $\boldsymbol{h}(\cdot)$ is parameterized by neural networks with $\boldsymbol{h} : \mathbb{R}^d \to \mathbb{R}^{n \times n}$ and $\boldsymbol{g}(\cdot)$ is the element-wise sigmoid function with $\boldsymbol{g} : \mathbb{R}^{n \times n} \to [0,1]^{n \times n}$. In particular, we use neural networks, transferring the latent variable $\boldsymbol{z}^t \in \mathbb{R}^d$ to $\boldsymbol{U}^t \in \mathbb{R}^{n \times k}$ and $\boldsymbol{V}^t \in \mathbb{R}^{n \times k}$. We let the latent dimensions $d$ and $k$ be smaller than the number of nodes $n$, and the outputs of neural networks are defined as

$$\boldsymbol{h}(\boldsymbol{z}^t) = \begin{cases} \boldsymbol{U}^t \boldsymbol{V}^{t\top} \in \mathbb{R}^{n \times n}, & \text{for directed network,} \\ \boldsymbol{U}^t \boldsymbol{U}^{t\top} \in \mathbb{R}^{n \times n}, & \text{for undirected network.} \end{cases} \tag{1}$$

Comparing to a decoder that directly outputs an $n$ by $n$ matrix, the decoder via matrix multiplication can reduce the number of neural network parameters. While this decoder focuses on homophily (the tendency for similar nodes to connect), an extension to consider heterophily (the tendency for dissimilar nodes to connect) as in Luan et al. (2022), Zhu et al. (2023), Di Francesco et al. (2024), and Luan et al. (2024) is allowed for future development.

Figure 1 gives an overview of the proposed framework, where graphs are sampled from latent variables in a top-down manner. Intuitively, the graph decoder can be helpful for learning graph-level representations in a bottom-up manner, compressing the enormous relations in $\boldsymbol{y}^t$ to extract the structural patterns through node-level representations $\boldsymbol{U}^t$ and $\boldsymbol{V}^t$ as an intermediary. The graph decoder $P_\phi(\boldsymbol{y}^t|\boldsymbol{z}^t)$, with neural network parameter $\boldsymbol{\phi}$, is shared across the time points $t = 1, \ldots, T$. It is also worth pointing out the simplicity of our framework, without the need of encoders.

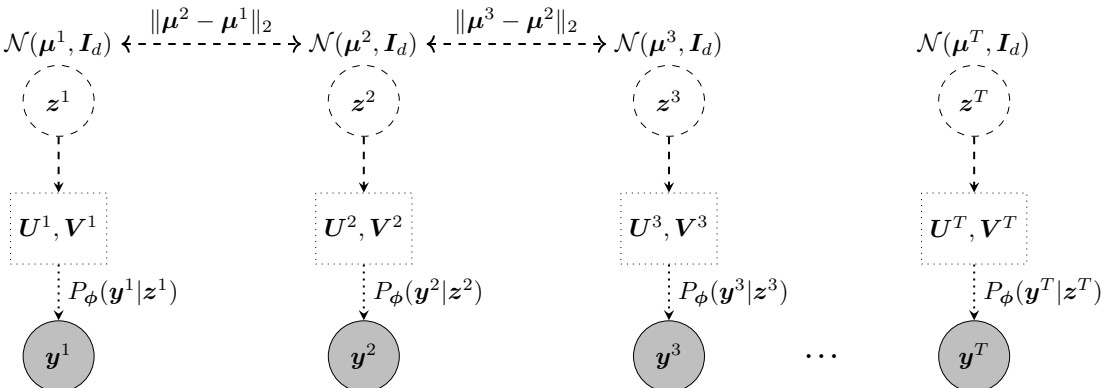

Figure 1: An overview of prior distributions and graph decoder for time-series of networks. The Group Fused Lasso regularization imposed on the sequential differences of prior parameters is elaborated in Section 3.

# 3 Learning and Inference

## 3.1 Learning Priors from Dynamic Graphs

Inspired by Vert & Bleakley (2010) and Bleakley & Vert (2011), we formulate the change point detection problem as a Group Fused Lasso problem (Alaíz et al., 2013). Denote the log-likelihood of the distribution for $\boldsymbol{y}^1, \ldots, \boldsymbol{y}^T$ as $l(\boldsymbol{\phi}, \boldsymbol{\mu})$. We want to solve

$$\hat{\boldsymbol{\phi}}, \hat{\boldsymbol{\mu}} = \operatorname*{arg\,min}_{\boldsymbol{\phi}, \boldsymbol{\mu}} -l(\boldsymbol{\phi}, \boldsymbol{\mu}) + \lambda \sum_{t=1}^{T-1} \|\boldsymbol{\mu}^{t+1} - \boldsymbol{\mu}^t\|_2 \tag{2}$$

where $\lambda > 0$ is a tuning parameter for the Group Fused Lasso penalty term.

The Group Fused Lasso penalty is useful for change point detection because it enforces piecewise constant patterns in the learned parameters, by minimizing the multivariate total variation. Specifically, the regularization term, expressed as the sum of the $\ell_2$ norms, encourages sparsity of the differences $\boldsymbol{\mu}^{t+1} - \boldsymbol{\mu}^t \in \mathbb{R}^d$, while allowing multiple coordinates across the $d$ dimensional differences to change at the same time $t$. The latter is often referred as a grouping effect that could not be achieved with the $\ell_1$ penalty of the differences. Furthermore, since the regularization is imposed on the prior parameters that relate to the likelihood of the data, the learned priors incorporate the structural changes from the observed graphs into the latent space. In summary, by penalizing the sum of sequential differences between the prior parameters, the proposed framework focuses on capturing meaningful structural changes while smoothing out minor variations.

Albeit the proposed framework in Section 2 is straightforward, parameter learning is challenging. To solve the optimization problem in (2) that involves latent variables, we need to manipulate the objective function accordingly. We first introduce a slack variable $\boldsymbol{\nu} \in \mathbb{R}^{T \times d}$ where $\boldsymbol{\nu}^t \in \mathbb{R}^d$ denotes the $t$-th row of matrix $\boldsymbol{\nu}$, and we rewrite the original problem as a constrained optimization problem:

$$\hat{\boldsymbol{\phi}}, \hat{\boldsymbol{\mu}} = \underset{\boldsymbol{\phi}, \boldsymbol{\mu}}{\arg\min} \, -l(\boldsymbol{\phi}, \boldsymbol{\mu}) + \lambda \sum_{t=1}^{T-1} \|\boldsymbol{\nu}^{t+1} - \boldsymbol{\nu}^t\|_2 \tag{3}$$
$$\text{subject to} \quad \boldsymbol{\mu} = \boldsymbol{\nu}.$$

Let $\boldsymbol{w} \in \mathbb{R}^{T \times d}$ be the scaled dual variable. The augmented Lagrangian can be expressed as

$$\mathcal{L}(\boldsymbol{\phi}, \boldsymbol{\mu}, \boldsymbol{\nu}, \boldsymbol{w}) = -l(\boldsymbol{\phi}, \boldsymbol{\mu}) + \lambda \sum_{t=1}^{T-1} \|\boldsymbol{\nu}^{t+1} - \boldsymbol{\nu}^t\|_2 + \frac{\kappa}{2}\|\boldsymbol{\mu} - \boldsymbol{\nu} + \boldsymbol{w}\|_F^2 - \frac{\kappa}{2}\|\boldsymbol{w}\|_F^2 \tag{4}$$

where $\kappa > 0$ is a penalty parameter for the augmentation term.

In practice, gradient descent may not work well for an objective function with Group Fused Lasso penalty. To this end, we introduce two more variables $(\boldsymbol{\gamma}, \boldsymbol{\beta}) \in \mathbb{R}^{1 \times d} \times \mathbb{R}^{(T-1) \times d}$ to ease the optimization, by converting it into a Group Lasso problem (Yuan & Lin, 2006). They are defined as

$$\boldsymbol{\gamma} = \boldsymbol{\nu}^1 \quad \text{and} \quad \boldsymbol{\beta}_{t,.} = \boldsymbol{\nu}^{t+1} - \boldsymbol{\nu}^t \ \ \forall \, t = 1, \ldots, T-1.$$

Reversely, the slack variable $\boldsymbol{\nu} \in \mathbb{R}^{T \times d}$ can be reconstructed as

$$\boldsymbol{\nu} = \mathbf{1}_{T,1}\boldsymbol{\gamma} + \boldsymbol{X}\boldsymbol{\beta}$$

where $\boldsymbol{X}$ is a $T \times (T-1)$ design matrix with $\boldsymbol{X}_{ij} = 1$ for $i > j$ and 0 otherwise. Substituting the $\boldsymbol{\nu}$ in (4) with $(\boldsymbol{\gamma}, \boldsymbol{\beta})$, the augmented Lagrangian is updated to

$$\mathcal{L}(\boldsymbol{\phi}, \boldsymbol{\mu}, \boldsymbol{\gamma}, \boldsymbol{\beta}, \boldsymbol{w}) = -l(\boldsymbol{\phi}, \boldsymbol{\mu}) + \lambda \sum_{t=1}^{T-1} \|\boldsymbol{\beta}_{t,.}\|_2 + \frac{\kappa}{2}\|\boldsymbol{\mu} - \mathbf{1}_{T,1}\boldsymbol{\gamma} - \boldsymbol{X}\boldsymbol{\beta} + \boldsymbol{w}\|_F^2 - \frac{\kappa}{2}\|\boldsymbol{w}\|_F^2. \tag{5}$$

Thus, we can derive the following Alternating Direction Method of Multipliers (ADMM) procedure (Boyd et al., 2011; Zhu, 2017; Wang et al., 2019) to solve the constrained optimization problem in (3):

$$\boldsymbol{\phi}_{(a+1)}, \boldsymbol{\mu}_{(a+1)} = \underset{\boldsymbol{\phi}, \boldsymbol{\mu}}{\arg\min} \, -l(\boldsymbol{\phi}, \boldsymbol{\mu}) + \frac{\kappa}{2}\|\boldsymbol{\mu} - \boldsymbol{\nu}_{(a)} + \boldsymbol{w}_{(a)}\|_F^2, \tag{6}$$

$$\boldsymbol{\gamma}_{(a+1)}, \boldsymbol{\beta}_{(a+1)} = \underset{\boldsymbol{\gamma}, \boldsymbol{\beta}}{\arg\min} \, \lambda \sum_{t=1}^{T-1} \|\boldsymbol{\beta}_{t,.}\|_2 + \frac{\kappa}{2}\|\boldsymbol{\mu}_{(a+1)} - \mathbf{1}_{T,1}\boldsymbol{\gamma} - \boldsymbol{X}\boldsymbol{\beta} + \boldsymbol{w}_{(a)}\|_F^2, \tag{7}$$

$$\boldsymbol{w}_{(a+1)} = \boldsymbol{\mu}_{(a+1)} - \boldsymbol{\nu}_{(a+1)} + \boldsymbol{w}_{(a)}, \tag{8}$$

where subscript $a$ denotes the current ADMM iteration. We recursively implement the three updates until certain convergence criterion is satisfied. Essentially, ADMM decomposes the optimization problem in (5) into smaller problems, solving each component with specific method derived in Section 3.2.

### 3.2 Parameters Update

#### 3.2.1 Updating $\boldsymbol{\mu}$ and $\boldsymbol{\phi}$

In this section, we derive the updates for the prior and graph decoder parameters. The prior parameters are inferred from the observed data as empirical Bayes. Denote the objective function in (6) as $\mathcal{L}(\boldsymbol{\phi}, \boldsymbol{\mu})$. Setting the gradients of $\mathcal{L}(\boldsymbol{\phi}, \boldsymbol{\mu})$ with respect to the prior parameter $\boldsymbol{\mu}^t \in \mathbb{R}^d$ to zeros, we have the following:

**Proposition 1.** *The solution for $\boldsymbol{\mu}^t$ at an iteration of our proposed ADMM procedure is a weighted sum:*

$$\boldsymbol{\mu}^t = \frac{1}{1+\kappa}\mathbb{E}_{P(\boldsymbol{z}^t|\boldsymbol{y}^t)}(\boldsymbol{z}^t) + \frac{\kappa}{1+\kappa}(\boldsymbol{\nu}^t - \boldsymbol{w}^t) \tag{9}$$

*between the conditional expectation of the latent variable under the posterior distribution $P(\boldsymbol{z}^t|\boldsymbol{y}^t)$ and the difference between the slack and the scaled dual variables. The term $\boldsymbol{w}^t \in \mathbb{R}^d$ denotes the $t$-th row of the scaled dual variable $\boldsymbol{w} \in \mathbb{R}^{T \times d}$. The derivation is provided in Appendix 7.1.*

Moreover, the gradient of $\mathcal{L}(\boldsymbol{\phi}, \boldsymbol{\mu})$ with respect to the graph decoder parameter $\boldsymbol{\phi}$ is calculated as

$$\nabla_{\boldsymbol{\phi}} \, \mathcal{L}(\boldsymbol{\phi}, \boldsymbol{\mu}) = -\sum_{t=1}^{T}\mathbb{E}_{P(\boldsymbol{z}^t|\boldsymbol{y}^t)}\Big(\nabla_{\boldsymbol{\phi}}\log P(\boldsymbol{y}^t|\boldsymbol{z}^t)\Big). \tag{10}$$

The parameter $\boldsymbol{\phi}$ can be updated efficiently through back-propagation.

Notably, calculating the solution in (9) and gradient in (10) requires evaluating the conditional expectation under the posterior distribution $P(\boldsymbol{z}^t|\boldsymbol{y}^t) \propto P(\boldsymbol{y}^t|\boldsymbol{z}^t) \times P(\boldsymbol{z}^t)$. We employ Langevin Dynamics, a short-run MCMC, to sample from the posterior distribution, approximating the conditional expectations (Xie et al., 2017; 2018; Nijkamp et al., 2020; Pang et al., 2020). In particular, let subscript $\tau$ be the time step of the Langevin Dynamics and let $\delta$ be a small step size. Moving toward the gradient of the posterior with respect to the latent variable, the Langevin Dynamics to draw samples from the posterior distribution is achieved by iterating the following:

$$\begin{aligned}
\boldsymbol{z}^t_{\tau+1} &= \boldsymbol{z}^t_\tau + \delta\big[\nabla_{\boldsymbol{z}^t}\log P(\boldsymbol{z}^t|\boldsymbol{y}^t)\big] + \sqrt{2\delta}\boldsymbol{\epsilon} \\
&= \boldsymbol{z}^t_\tau + \delta\big[\nabla_{\boldsymbol{z}^t}\log P_{\boldsymbol{\phi}}(\boldsymbol{y}^t|\boldsymbol{z}^t) - (\boldsymbol{z}^t_\tau - \boldsymbol{\mu}^t)\big] + \sqrt{2\delta}\boldsymbol{\epsilon}
\end{aligned} \tag{11}$$

where $\boldsymbol{\epsilon} \sim \mathcal{N}(\boldsymbol{0}, \boldsymbol{I}_d)$ is a random perturbation to the process. The derivation is provided in Appendix 7.2. Different from the VAE framework where latent variables are obtained through an encoder, we sampled the latent variables from the posterior distributions via Langevin Dynamics.

#### 3.2.2 Updating $\boldsymbol{\gamma}$ and $\boldsymbol{\beta}$

In this section, we derive the update in (7), which is equivalent to solving a Group Lasso problem. The grouping effect allows the $d$ dimensional differences to change at the same time $t$. With ADMM, the updates on $\boldsymbol{\gamma}$ and $\boldsymbol{\beta}$ do not require the observed network data $\{\boldsymbol{y}^t\}_{t=1}^T$. By adapting the derivation in Bleakley & Vert (2011), we have the following:

**Proposition 2. [Bleakley & Vert, 2011]** *The Group Lasso problem to update $\boldsymbol{\beta} \in \mathbb{R}^{(T-1)\times d}$ is solved in a block coordinate descent manner, by iteratively applying the following equation to each row $t$:*

$$\boldsymbol{\beta}_{t,\cdot} \leftarrow \frac{1}{\kappa\boldsymbol{X}_{\cdot,t}^{\top}\boldsymbol{X}_{\cdot,t}}\left(1 - \frac{\lambda}{\|\boldsymbol{b}_t\|_2}\right)_{+}\boldsymbol{b}_t \tag{12}$$

*where $(\cdot)_{+} = \max(\cdot, 0)$ and*

$$\boldsymbol{b}_t = \kappa\boldsymbol{X}_{\cdot,t}^{\top}(\boldsymbol{\mu}_{(a+1)} + \boldsymbol{w}_{(a)} - \boldsymbol{1}_{T,1}\boldsymbol{\gamma} - \boldsymbol{X}_{\cdot,-t}\boldsymbol{\beta}_{-t,\cdot}).$$

*The derivation is provided in Appendix 7.3.*

In particular, $\boldsymbol{\beta}_{t,\cdot} \in \mathbb{R}^d$ becomes $\mathbf{0}$ when $\|\boldsymbol{b}_t\|_2 \leq \lambda$. Also, the convergence of the procedure can be monitored by the Karush-Kuhn-Tucker (KKT) conditions:

$$\lambda \frac{\boldsymbol{\beta}_{t,\cdot}}{\|\boldsymbol{\beta}_{t,\cdot}\|_2} - \kappa \boldsymbol{X}_{\cdot,t}^\top (\boldsymbol{\mu}_{(a+1)} + \boldsymbol{w}_{(a)} - \mathbf{1}_{T,1}\boldsymbol{\gamma} - \boldsymbol{X}\boldsymbol{\beta}) = \mathbf{0} \qquad \forall \boldsymbol{\beta}_{t,\cdot} \neq \mathbf{0},$$

$$\|{-\kappa}\boldsymbol{X}_{\cdot,t}^\top (\boldsymbol{\mu}_{(a+1)} + \boldsymbol{w}_{(a)} - \mathbf{1}_{T,1}\boldsymbol{\gamma} - \boldsymbol{X}\boldsymbol{\beta})\|_2 \leq \lambda \qquad \forall \boldsymbol{\beta}_{t,\cdot} = \mathbf{0}.$$

Lastly, the minimum in $\boldsymbol{\gamma} \in \mathbb{R}^{1 \times d}$ is achieved at

$$\boldsymbol{\gamma} = (1/T)\mathbf{1}_{1,T} \cdot (\boldsymbol{\mu}_{(a+1)} + \boldsymbol{w}_{(a)} - \boldsymbol{X}\boldsymbol{\beta}).$$

The algorithm for the ADMM procedure is provided in Appendix 7.4 and details about the implementation are provided in Appendix 7.5.

## 4 Change Point Localization and Model Selection

### 4.1 Change Point Localization

In this section, we provide two effective methods to localize the change points after parameter learning, and they can be used for different purposes. For the first approach, we resort to the prior distribution where $\boldsymbol{z}^t \sim \mathcal{N}(\boldsymbol{\mu}^t, \boldsymbol{I}_d)$. When no change occurs or $\boldsymbol{\mu}^t - \boldsymbol{\mu}^{t-1} = \mathbf{0}$, we have $\boldsymbol{z}^t - \boldsymbol{z}^{t-1} \sim \mathcal{N}(\mathbf{0}, 2\boldsymbol{I}_d)$ and

$$u^t := \frac{1}{2}(\boldsymbol{z}^t - \boldsymbol{z}^{t-1})^\top (\boldsymbol{z}^t - \boldsymbol{z}^{t-1}) \sim \chi_d^2.$$

Furthermore, the mean of $u^t$ over $m$ samples follows a Gamma distribution:

$$\bar{u}_m^t \sim \Gamma(\theta = \frac{2}{m}, \xi = \frac{md}{2})$$

where $\theta$ and $\xi$ are the respective scale and shape parameters.

As we capture the structural changes in the latent space, we can draw samples from the learned priors to reflect the sequential changes. In particular, for a time point $t$, we sample $\hat{\boldsymbol{z}}^t - \hat{\boldsymbol{z}}^{t-1}$ from $\mathcal{N}(\hat{\boldsymbol{\mu}}^t - \hat{\boldsymbol{\mu}}^{t-1}, 2\boldsymbol{I}_d)$, and we perform the same transformation:

$$v^t := \frac{1}{2}(\hat{\boldsymbol{z}}^t - \hat{\boldsymbol{z}}^{t-1})^\top (\hat{\boldsymbol{z}}^t - \hat{\boldsymbol{z}}^{t-1}).$$

Then we compare the mean of $v^t$ over $m$ samples with a quantile:

$$\mathbb{P}(\bar{v}_m^t > q_{\text{thr}}) = 1 - \frac{\alpha}{T-1} \tag{13}$$

where $q_{\text{thr}}$ is the $1 - \alpha/(T-1)$ quantile of the Gamma distribution for $\bar{u}_m^t$ when no change occurs. We consider the time point $t$ with $\bar{v}_m^t > q_{\text{thr}}$ as the detected change point.

For the second approach, we can directly utilize the localizing method from Kei et al. (2023b), which is more robust in practice, as compared in the simulation study of Section 5.1. First, we calculate the differences between consecutive time points in $\hat{\boldsymbol{\mu}} \in \mathbb{R}^{T \times d}$ as

$$\Delta\hat{\boldsymbol{\mu}}^t = \|\boldsymbol{\mu}^t - \boldsymbol{\mu}^{t-1}\|_2 \quad \forall\, t \in [2, T].$$

Then we standardize the differences as

$$\Delta\hat{\boldsymbol{\zeta}}^t = \frac{\Delta\hat{\boldsymbol{\mu}}^t - \text{median}(\Delta\hat{\boldsymbol{\mu}})}{\text{std}(\Delta\hat{\boldsymbol{\mu}})} \quad \forall\, t \in [2, T] \tag{14}$$

and construct a data-driven threshold defined as

$$\mathcal{T}_{\text{thr}} := \text{mean}(\Delta\hat{\boldsymbol{\zeta}}) + \mathcal{Z}_q \times \text{std}(\Delta\hat{\boldsymbol{\zeta}}) \tag{15}$$

where $\mathcal{Z}_q$ is the $q\%$ quantile of the standard Gaussian distribution $\mathcal{N}(0, 1)$. We declare a change point $C_k$ when $\Delta\hat{\zeta}^{C_k} > \mathcal{T}_{\text{thr}}$.

The data-driven threshold in (15) is intuitive, as the standardized differences $\Delta\hat{\zeta}$ between two consecutive change points are close to zeros, while the differences that are at the change points are substantially greater than zeros. When traced in a plot over time $t$, the $\Delta\hat{\zeta}$ can exhibit the magnitude of structural changes, and the threshold that deviates from the mean provides a reasonable cut-off value for the standardized differences, as demonstrated in Figures 11 and 12. In summary, the localizing method derived from the prior distribution has a statistical justification, while the localizing method with the data-driven threshold is more robust for different types of network data in practice.

## 4.2 Model Selection

The optimization problem in (3) involves a tuning parameter that can yield different sets of detected change points when it is varied. In this work, we use Cross-Validation to select $\lambda$. In particular, we split the original time series of graphs into training and testing sets: the training set consists of graphs at odd indexed time points and the testing set consists of graphs at even indexed time points. Fixed on a specific $\lambda$ value, we learn the model parameters with the training set, and we evaluate the learned model with the testing set.

For a list of $\lambda$ values, we choose the $\lambda$ giving the maximal log-likelihood on the testing set. Note that the log-likelihood is approximated by Monte Carlo samples $\{z_u^t\}_{u=1}^s$ drawn from the prior distribution $P(z^t)$ as

$$\sum_{t=1}^{T} \log P(\boldsymbol{y}^t) \approx \sum_{t=1}^{T} \log \left[ \frac{1}{s} \sum_{u=1}^{s} \Big[ \prod_{(i,j) \in \mathbb{Y}} P_\phi(\boldsymbol{y}_{ij}^t | \boldsymbol{z}_u^t) \Big] \right].$$

Further computational details are discussed in Appendix 7.5. With the selected $\lambda$ value, we learn the model parameters again with the full data, resulting the final set of detected change points.

# 5 Simulated and Real Data Experiments

## 5.1 Simulation Study

In this section, we implement the proposed method on simulated data. To evaluate the performance of change point detection, we use three standard metrics in the literature that focus on the number of change points, the time gap between the true and detected change points, and the coverage over the segmented time intervals. The first metric is the absolute error $|\hat{K} - K|$ where $\hat{K}$ and $K$ are the respective numbers of the detected and true change points. The second metric described in Madrid Padilla et al. (2021) is the one-sided Hausdorff distance, which is defined as

$$d(\hat{\mathcal{C}}|\mathcal{C}) = \max_{c \in \mathcal{C}} \min_{\hat{c} \in \hat{\mathcal{C}}} |\hat{c} - c|$$

where $\hat{\mathcal{C}}$ and $\mathcal{C}$ are the respective sets of detected and true change points. Also, we report the reversed one-sided Hausdorff distance $d(\mathcal{C}|\hat{\mathcal{C}})$. By convention, when $\hat{\mathcal{C}} = \emptyset$, we let $d(\hat{\mathcal{C}}|\mathcal{C}) = \infty$ and $d(\mathcal{C}|\hat{\mathcal{C}}) = -\infty$. The last metric described in van den Burg & Williams (2020) is the coverage of a partition $\mathcal{G}$ by another partition $\mathcal{G}'$, which is defined as

$$C(\mathcal{G}, \mathcal{G}') = \frac{1}{T} \sum_{\mathcal{A} \in \mathcal{G}} |\mathcal{A}| \cdot \max_{\mathcal{A}' \in \mathcal{G}'} \frac{|\mathcal{A} \cap \mathcal{A}'|}{|\mathcal{A} \cup \mathcal{A}'|}$$

with $\mathcal{A}, \mathcal{A}' \subseteq [1, T]$. The $\mathcal{G}$ and $\mathcal{G}'$ are collections of intervals between consecutive change points for the respective ground truth and detected results.

We simulate dynamic graphs from three scenarios to compare the performance of the proposed and competitor methods: Separable Temporal Exponential Random Graph Model, Stochastic Block Model, and Recurrent Neural Networks. For each scenario with different numbers of nodes $n \in \{50, 100\}$, we simulate 10 Monte

Carlo trials of directed networks with time span $T = 100$. The true change points are located at $t = \{26, 51, 76\}$, so the number of change points $K = 3$. Moreover, the $K + 1 = 4$ intervals in the partition $\mathcal{G}$ are $\mathcal{A}_1 = \{1, \ldots, 25\}$, $\mathcal{A}_2 = \{26, \ldots, 50\}$, $\mathcal{A}_3 = \{51, \ldots, 75\}$, and $\mathcal{A}_4 = \{76, \ldots, 100\}$. In each specification, we report the means and standard deviations over 10 Monte Carlo trials for the evaluation metrics. CPDlatent$_N$ denotes our proposed approach with the data-driven threshold in (15), using 90% quantile from standard Normal distribution. We let the latent dimensions $d = 10$ and $k = 5$ for the graph decoder. CPDlatent$_G$ denotes our proposed approach with the localizing method in (13), using $\alpha = 0.01$ from Gamma distribution. We let the latent dimensions $d = n/10$ and $k = 10$ for the graph decoder. The number of samples drawn from the Gamma distribution is $m = 1000$ when $d = 5$ and $m = 500$ when $d = 10$.

Five competitors, gSeg (Chen & Zhang, 2015), kerSeg (Song & Chen, 2022b), CPDrdpg (Madrid Padilla et al., 2022), CPDnbs (Wang et al., 2021a), and CPDstergm (Kei et al., 2023b), are provided for comparison. The gSeg method utilizes a graph-based scan statistics and the kerSeg method employs a kernel-based framework to test the partition before and after a potential change point. The CPDrdpg method detects change points by estimating the latent positions from a RDPG model and by constructing a non-parametric CUSUM statistic that allows for temporal dependence. The CPDnbs method detects change points by combining sample splitting with wild binary segmentation (WBS) and by maximizing the inner product of two CUSUM statistics computed from the samples. The CPDstergm method fits a STERGM with user-specified network statistics to detect change points based on the total variation of estimated parameters.

For CPDstergm, we first use two network statistics, edge count and mutuality, in both formation and dissolution models to let $p = 4$. We then add one more network statistic, number of triangles, to let $p = 6$ as another specification. For CPDrdpg, we let the number of intervals for WBS be $W = 50$, and we let the number of leading singular values of an adjacency matrix in the scaled PCA algorithm be $d = 5$. For CPDnbs, we let the number of intervals for WBS be $W = 15$ and we set the threshold to the order of $n \log^2(T)$. For kerSeg, we use the approximated p-value of fGKCP$_1$, and we set $\alpha = 0.001$. For gSeg, we use the minimum spanning tree to construct the similarity graph, with the approximated p-value of the original edge count scan statistic, and we set $\alpha = 0.05$. Moreover, as gSeg and kerSeg are general methods for change point detection, we use networks (nets.) and network statistics (stats.) as two types of input data. Throughout the paper, we choose these settings because they produce good performance on average for the competitors. Changing these settings can enhance their performance on some specifications, while severely jeopardizing their performance on other specifications.

**Scenario 1: Separable Temporal Exponential Random Graph Model**

In this scenario, we apply time-homogeneous Separable Temporal Exponential Random Graph Model (STERGM) between change points to simulate sequences of dynamic networks (Krivitsky & Handcock, 2014). We use three network statistics, edge count, mutuality, and number of triangles, in both formation (F) and dissolution (D) models. The $p = 6$ parameters for each time point $t$ are

$$\boldsymbol{\theta}_F^t, \boldsymbol{\theta}_D^t = \begin{cases} -2, \ 2, \ -2, \ -1, \ 2, \ 1, & t \in \mathcal{A}_1 \cup \mathcal{A}_3 \setminus 1, \\ -1.5, \ 1, \ -1, \ 2, \ 1, \ 1.5, & t \in \mathcal{A}_2 \cup \mathcal{A}_4. \end{cases}$$

Figure 2 exhibits examples of simulated networks. Visually, STERGM produces adjacency matrices that are sparse, which is often the case in real world social networks.

Table 1 displays the means and standard deviations of the evaluation metrics for comparison. Since the dynamic networks are directly sampled from STERGM, the CPDstergm method with correctly specified network statistics ($p = 6$) achieves the best performance, in terms of greater converge of time intervals. However, when the network statistics are mis-specified with $p = 4$, the performance of CPDstergm is substantially worsened, with greater time gaps between the true and detected change points. The deteriorating performance of CPDstergm emphasizes the importance of graph-level features in change point detection. Similarly, while the CPDrdpg method detects the correct numbers of change points on average, the time gaps between the true and detected change points are large. Similarly, the detected change points from the CPDnbs method also have greater time gaps. Moreover, using either networks (nets.) or network statistics (stats.) cannot improve the performance of gSeg and kerSeg methods. The binary segmentation approach

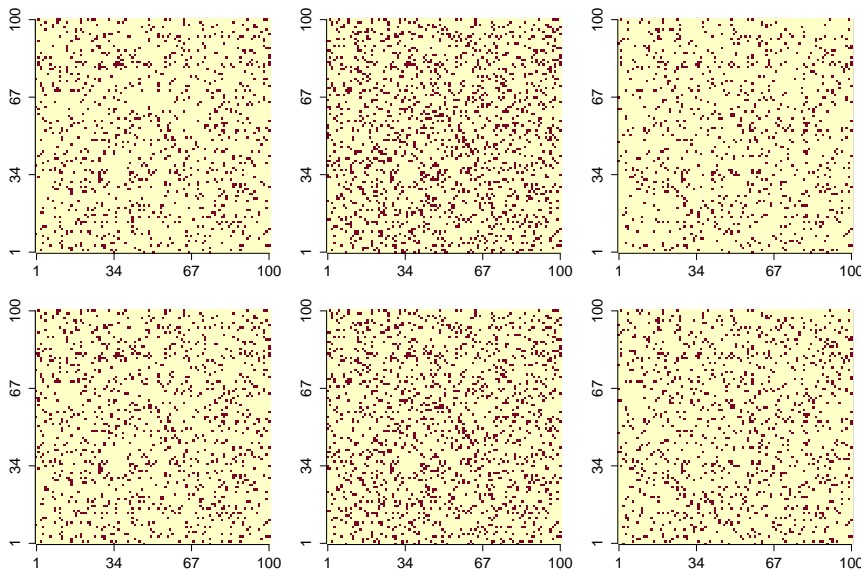

Figure 2: Examples of networks simulated from STERGM with number of nodes $n = 100$. The edge density is approximately 15% for each network. In the first row, from left to right, each plot corresponds to the network at $t = 25, 50, 75$ respectively. In the second row, from left to right, each plot corresponds to the network at $t = 26, 51, 76$ respectively (the change points).

tends to detect excessive numbers of change points, capturing noises from the data. In this scenario, although the CPDstergm method with $p = 6$ achieves the best performance, the true network statistics are usually not known to the modeler a priori. Our CPDlatent method, without the need of specifying network statistics, can achieve good performance on average.

**Scenario 2: Stochastic Block Model**

In this scenario, we use Stochastic Block Model (SBM) to simulate sequences of dynamic networks, and we impose a time-dependent mechanism in the simulation process as in Madrid Padilla et al. (2022). Two probability matrices $\boldsymbol{P}, \boldsymbol{Q} \in [0, 1]^{n \times n}$ are constructed and they are defined as

$$\boldsymbol{P}_{ij} = \begin{cases} 0.5, & i, j \in \mathcal{B}_l, \ l \in [3], \\ 0.3, & \text{otherwise,} \end{cases} \quad \text{and} \quad \boldsymbol{Q}_{ij} = \begin{cases} 0.45, & i, j \in \mathcal{B}_l, \ l \in [3], \\ 0.2, & \text{otherwise,} \end{cases}$$

where $\mathcal{B}_1, \mathcal{B}_2, \mathcal{B}_3$ are evenly sized clusters that form a partition of $\{1, \ldots, n\}$. Then a sequence of matrices $\boldsymbol{E}^t \in [0, 1]^{n \times n}$ are arranged for $t = 1, \ldots, T$ such that

$$\boldsymbol{E}_{ij}^t = \begin{cases} \boldsymbol{P}_{ij}, & t \in \mathcal{A}_1 \cup \mathcal{A}_3, \\ \boldsymbol{Q}_{ij}, & t \in \mathcal{A}_2 \cup \mathcal{A}_4. \end{cases}$$

Lastly, the networks are simulated with $\rho = 0.5$ as the time-dependent mechanism. For $t = 1, \ldots, T-1$, we let $\boldsymbol{y}_{ij}^1 \sim \text{Bernoulli}(\boldsymbol{E}_{ij}^1)$ and

$$\boldsymbol{y}_{ij}^{t+1} \sim \begin{cases} \text{Bernoulli}\big(\rho(1 - \boldsymbol{E}_{ij}^{t+1}) + \boldsymbol{E}_{ij}^{t+1}\big), & \boldsymbol{y}_{ij}^t = 1, \\ \text{Bernoulli}\big((1 - \rho)\boldsymbol{E}_{ij}^{t+1}\big), & \boldsymbol{y}_{ij}^t = 0. \end{cases}$$

With $\rho > 0$, the probability to form an edge for $i, j$ becomes greater at time $t + 1$ when there exists an edge at time $t$, and the probability becomes smaller when there does not exist an edge at time $t$. Figure 3 exhibits

Table 1: Means (standard deviations) of evaluation metrics for dynamic graphs simulated from STERGM. The best coverage metric is bolded.

| $n$ | Method | $|\hat{K} - K| \downarrow$ | $d(\hat{\mathcal{C}}|\mathcal{C}) \downarrow$ | $d(\mathcal{C}|\hat{\mathcal{C}}) \downarrow$ | $C(\mathcal{G}, \mathcal{G}') \uparrow$ |
|---|---|---|---|---|---|
| | $\text{CPDlatent}_N$ | 0.1 (0.3) | 4.3 (5.7) | 2.6 (1.3) | 90.87% |
| | $\text{CPDlatent}_G$ | 0.4 (0.6) | 4.2 (6.9) | 3.4 (3.4) | 90.97% |
| | CPDrdpg | 0.9 (1.6) | 8.7 (9.5) | 8.5 (6.0) | 76.27% |
| | CPDnbs | 1.2 (0.6) | 4.6 (3.9) | 11.0 (0.9) | 75.80% |
| 50 | $\text{CPDstergm}_{p=4}$ | 1.5 (0.8) | 11.7 (7.5) | 10.5 (2.3) | 67.68% |
| | $\text{CPDstergm}_{p=6}$ | 0.2 (0.4) | 1.6 (1.2) | 3 (3.5) | **91.54**% |
| | gSeg (nets.) | 12.3 (0.5) | 0 (0.0) | 19 (0.0) | 27.90% |
| | gSeg (stats.) | 15.8 (0.7) | 1.5 (0.5) | 20.1 (0.3) | 24.55% |
| | kerSeg (nets.) | 9.7 (0.9) | 1.4 (0.9) | 17.9 (1.2) | 37.62% |
| | kerSeg (stats.) | 9.4 (0.7) | 3.9 (1.3) | 18 (1.8) | 35.86% |
| | $\text{CPDlatent}_N$ | 0 (0.0) | 3.9 (1.3) | 3.9 (1.3) | 91.33% |
| | $\text{CPDlatent}_G$ | 0.7 (1.3) | 3.1 (1.3) | 6.0 (4.0) | 88.55% |
| | CPDrdpg | 0.8 (1.0) | 4.5 (2.0) | 8.2 (4.7) | 80.54% |
| | CPDnbs | 1.4 (0.5) | 4.9 (3.7) | 11.0 (0.9) | 72.99% |
| 100 | $\text{CPDstergm}_{p=4}$ | 0.7 (0.6) | 21.9 (10.3) | 7.6 (4.3) | 67.21% |
| | $\text{CPDstergm}_{p=6}$ | 0 (0.0) | 1.1 (0.3) | 1.1 (0.3) | **94.01**% |
| | gSeg (nets.) | 12 (0.0) | 0 (0.0) | 19 (0.0) | 28.00% |
| | gSeg (stats.) | 14.5 (2.3) | 3.3 (3.6) | 20.2 (0.4) | 26.13% |
| | kerSeg (nets.) | 9.3 (0.8) | 1 (0.0) | 17.7 (0.6) | 37.62% |
| | kerSeg (stats.) | 8.5 (0.8) | 4.5 (1.4) | 17.3 (1.7) | 36.92% |

Figure 3: Examples of networks simulated from SBM with number of nodes $n = 100$. The edge density is approximately 30% for each network. In the first row, from left to right, each plot corresponds to the network at $t = 25, 50, 75$ respectively. In the second row, from left to right, each plot corresponds to the network at $t = 26, 51, 76$ respectively (the change points).

examples of simulated networks. Visually, SBM produces adjacency matrices with block structures, where mutuality serves as an important pattern for the homophily within communities.

Table 2 displays the means and standard deviations of the evaluation metrics for comparison. As expected, both CPDstergm methods with $p = 4$ and $p = 6$ that utilize mutuality as a network statistic for detection achieve good performance, in terms of greater converge of time intervals. The CPDrdpg method also produce relatively good performance, but the time gaps between the true and detected change points are large. Similarly, the CPDnbs method also produces change points that have greater time gaps from the ground truth. Furthermore, using network statistics (stats.) with mutuality included for both gSeg and kerSeg methods improve their performance substantially, comparing to using networks (nets.) as input for detection. This again emphasizes the significance of using graph-level representation in change point detection for dynamic networks. Lastly, our CPDlatent method, which infers the features in the latent space that induce the structural changes, achieves the best performance in this scenario.

Table 2: Means (standard deviations) of evaluation metrics for dynamic networks simulated from SBM. The best coverage metric is bolded.

| $n$ | Method | $\|\hat{K} - K\| \downarrow$ | $d(\hat{\mathcal{C}}|\mathcal{C}) \downarrow$ | $d(\mathcal{C}|\hat{\mathcal{C}}) \downarrow$ | $C(\mathcal{G}, \mathcal{G}') \uparrow$ |
|---|---|---|---|---|---|
| | CPDlatent$_N$ | 0 (0.0) | 0.1 (0.3) | 0.1 (0.3) | **99.80**% |
| | CPDlatent$_G$ | 0.3 (0.6) | 0.1 (0.3) | 3.1 (6.2) | 96.70% |
| | CPDrdpg | 1.4 (1.8) | 2.2 (1.2) | 8.2 (6.0) | 81.01% |
| | CPDnbs | 1.6 (0.5) | 3.8 (3.6) | 11.4 (1.2) | 73.13% |
| 50 | CPDstergm$_{p=4}$ | 0.1 (0.3) | 1 (0.0) | 2.4 (4.2) | 97.04% |
| | CPDstergm$_{p=6}$ | 0.3 (0.5) | 1 (0.0) | 4.6 (5.6) | 94.74% |
| | gSeg (nets.) | 12.9 (1.8) | 0 (0.0) | 19.4 (0.8) | 27.20% |
| | gSeg (stats.) | 2.2 (0.7) | Inf (na) | $-$Inf (na) | 49.21% |
| | kerSeg (nets.) | 6.4 (1.4) | 0 (0.0) | 16.6 (2.0) | 45.50% |
| | kerSeg (stats.) | 0.9 (1.2) | 0 (0.0) | 5.6 (6.8) | 93.50% |
| | CPDlatent$_N$ | 0.1 (0.3) | 0.1 (0.3) | 1.3 (3.6) | **98.60**% |
| | CPDlatent$_G$ | 0.5 (0.7) | 0.2 (0.4) | 5.1 (6.1) | 94.81% |
| | CPDrdpg | 0.3 (0.6) | 1.5 (0.5) | 2.5 (2.0) | 91.05% |
| | CPDnbs | 1.8 (0.6) | 3.5 (3.3) | 12.3 (1.3) | 72.04% |
| 100 | CPDstergm$_{p=4}$ | 0 (0.0) | 1 (0.0) | 1 (0.0) | 98.04% |
| | CPDstergm$_{p=6}$ | 0 (0.0) | 1 (0.0) | 1 (0.0) | 98.04% |
| | gSeg (nets.) | 12.3 (0.9) | 0 (0.0) | 19 (0.0) | 27.80% |
| | gSeg (stats.) | 2 (0.4) | Inf (na) | $-$Inf (na) | 55.75% |
| | kerSeg (nets.) | 6 (0.8) | 0 (0.0) | 15.2 (2.0) | 47.00% |
| | kerSeg (stats.) | 0.9 (0.7) | 0 (0.0) | 9.6 (7.6) | 93.40% |

**Scenario 3: Recurrent Neural Networks**

In this scenario, we use Recurrent Neural Networks (RNNs) to simulate sequences of dynamic networks. Specifically, we sample latent variables $\boldsymbol{z}^t$ from pre-defined priors, and we randomly initialize the RNNs with uniform weights. The graphs are then generated by the matrix multiplication defined by Equation (1), using the outputs $\boldsymbol{U}^t$ and $\boldsymbol{V}^t$ from RNNs. The parameters for the pre-defined priors are

$$\boldsymbol{z}^t \sim \begin{cases} \mathcal{N}(-\mathbf{1}, \ 0.1\boldsymbol{I}_d), & t \in \mathcal{A}_1 \cup \mathcal{A}_3, \\ \mathcal{N}(\mathbf{5}, \ 0.1\boldsymbol{I}_d), & t \in \mathcal{A}_2 \cup \mathcal{A}_4. \end{cases}$$

Similar to the previous two scenarios, the simulation using RNNs also imposes a time-dependent mechanism across the dynamic networks. Figure 4 exhibits examples of simulated networks. Visually, RNNs produce adjacency matrices that are dense, and no discernible pattern can be noticed.

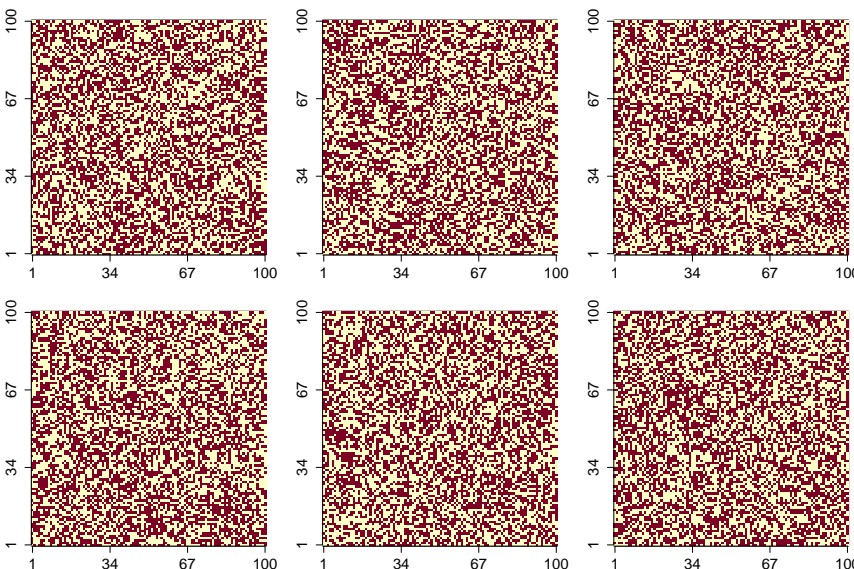

Figure 4: Examples of networks generated from RNNs with number of nodes $n = 100$. The edge density is approximately 50% for each network. In the first row, from left to right, each plot corresponds to the network at $t = 25, 50, 75$ respectively. In the second row, from left to right, each plot corresponds to the network at $t = 26, 51, 76$ respectively (the change points).

Table 3 displays the means and standard deviations of the evaluation metrics for comparison. Because no significant structural pattern or suitable network statistics can be determined a priori, neither CPDstergm method with $p = 4$ nor with $p = 6$ can detect the change points accurately. Likewise, both gSeg and kerSeg methods that utilize the mis-specified network statistics (stats.) cannot produce satisfactory performance. The CPDrdpg method that focuses on node-level representation also does not perform well for networks with complex structures. Notably, the kerSeg method that exploits the features in high dimension with networks (nets.) as input data can produce good performance. The CPDnbs method that assumes the networks are generated from inhomogeneous Bernoulli models and uses weighted averages of adjacency matrices also achieve good performance. Lastly, our CPDlatent method that first infers the graph-level representations from the networks and then utilizes them to detect change points yields the best performance in this scenario.

## 5.2 Degree Distributions and Shared Partner Distributions Comparison

Besides the ability to detect change points, our proposed framework includes a decoder that can sample graphs from latent variables. Consider the originally simulated networks as ground truth. To evaluate the model's goodness of fit, we compare the degree distributions and shared partner distributions (Hunter & Handcock, 2006) between the generated graphs and ground truth, as in Hunter et al. (2008a), Hunter et al. (2008b), Kolaczyk & Csárdi (2014) and Handcock et al. (2022). Specifically, we first estimate the model parameters using the simulated data that excludes the graphs at time $t \in \{10, 20, \ldots, 100\}$. Then we sample $\boldsymbol{z}^{t-1}$ from the estimated priors $\mathcal{N}(\hat{\boldsymbol{\mu}}^{t-1}, \boldsymbol{I}_d)$ to generate $\hat{\boldsymbol{y}}^{t-1}$ with the learned decoder, as out-of-sample forecasts for the networks at $t \in \{10, 20, \ldots, 100\}$. For each time point, we generate $s = 200$ networks and we visualize the degree and shared partners distributions. If the corresponding distributions are similar, it indicates that the graph decoder effectively captures the underlying structures, and the generated graphs closely resemble the originally simulated graphs. Figures 5, 6, and 7 display the degree distributions and Figures 8, 9, and 10 display the shared partner distributions of the generated graphs predicted from the learned decoder for Scenarios 1, 2, and 3 respectively.

Table 3: Means (standard deviations) of evaluation metrics for dynamic networks simulated from RNNs. The best coverage metric is bolded.

| $n$ | Method | $\|\hat{K} - K\| \downarrow$ | $d(\hat{\mathcal{C}}\|\mathcal{C}) \downarrow$ | $d(\mathcal{C}\|\hat{\mathcal{C}}) \downarrow$ | $C(\mathcal{G}, \mathcal{G}') \uparrow$ |
|---|---|---|---|---|---|
| 50 | $\text{CPDlatent}_N$ | 0 (0.0) | 1.8 (0.7) | 1.8 (0.7) | **94.77**% |
| | $\text{CPDlatent}_G$ | 0.3 (0.6) | 1.7 (0.6) | 3.2 (3.0) | 93.04% |
| | CPDrdpg | 2.4 (1.6) | 12.7 (7.5) | 11.2 (5.3) | 58.07% |
| | CPDnbs | 0.1 (0.3) | 3.5 (6.8) | 1.2 (0.4) | 93.87% |
| | $\text{CPDstergm}_{p=4}$ | 2.0 (1.7) | 6.0 (7.7) | 15.2 (4.9) | 72.10% |
| | $\text{CPDstergm}_{p=6}$ | 1.0 (0.4) | 18.5 (9.4) | 14.3 (2.9) | 60.25% |
| | gSeg (nets.) | 2.3 (0.6) | Inf (na) | $-$Inf (na) | 29.42% |
| | gSeg (stats.) | 2.9 (0.3) | Inf (na) | $-$Inf (na) | 2.47% |
| | kerSeg (nets.) | 1.5 (0.9) | 1.4 (0.7) | 5.3 (3.3) | 89.25% |
| | kerSeg (stats.) | 2.8 (0.4) | Inf (na) | $-$Inf (na) | 9.89% |
| 100 | $\text{CPDlatent}_N$ | 0 (0.0) | 2.5 (0.7) | 2.5 (0.7) | 91.96% |
| | $\text{CPDlatent}_G$ | 0.2 (0.6) | 2.1 (0.7) | 2.8 (1.8) | **92.34**% |
| | CPDrdpg | 1.5 (1.0) | 12.3 (8.2) | 10.4 (3.7) | 60.15% |
| | CPDnbs | 0.1 (0.3) | 2.5 (2.5) | 3.3 (3.4) | 90.74% |
| | $\text{CPDstergm}_{p=4}$ | 2.0 (1.4) | 10.6 (8.0) | 14.1 (3.1) | 60.37% |
| | $\text{CPDstergm}_{p=6}$ | 1.2 (1.3) | 20.6 (12.6) | 15.2 (5.9) | 53.21% |
| | gSeg (nets.) | 3 (0.0) | Inf (na) | $-$Inf (na) | 0% |
| | gSeg (stats.) | 2.9 (0.3) | Inf (na) | $-$Inf (na) | 4.27% |
| | kerSeg (nets.) | 1.4 (0.7) | 1.9 (0.7) | 5.4 (1.9) | 88.95% |
| | kerSeg (stats.) | 3 (0.0) | Inf (na) | $-$Inf (na) | 0% |

Since the networks simulated from STERGM are sparse, the node degrees for both simulated and generated graphs are low in Figure 5. The sparsity challenges the decoder to capture the structural patterns, which explain why some peaks in the degree distributions are not fully covered by the generated graphs. Nevertheless, for nodes with low degrees on the left tails, the trends are well captured by the decoder. Similarly, due to the sparsity, the majority of edges has fewer shared partners in Figure 8. Though the generated networks tends to over-estimate the numbers of edges with particular numbers of shared partners, the decreasing trends are captured by the decoder.

Next, the networks simulated from SBM have strong inter-block interactions and the networks simulated from RNNs are dense, resulting in higher node degrees for the simulated and generated graphs in Figures 6 and 7. For these two scenarios, both the tails and peaks of the degree distributions are fully covered by the generated graphs. Similarly, the numbers of shared partners for edges have wider ranges in Figures 9 and 10, and the overall trends are well captured by the decoder. For all three scenarios, the slight discrepancy in the alignment may be due to the decoder being shared across the time points, balancing the structural variation over time. In summary, the overall trends of the corresponding distributions for the generated graphs align with those for the simulated graphs, suggesting the generated graphs are similar to the ground truth and the learned decoders have captured the underlying graph structures.

### 5.3 Real Data Experiments

In this section, we apply the proposed method to two real data, and we align the detected change points with significant events for interpretation. In practice, the number and location of change points for real data are typically unknown, so there is no widely accepted ground truth for either the change points or their corresponding events in this unsupervised learning problem. Besides validating the detected change points with significant events as in the literature, we attempt a heuristic approach to compare the results across

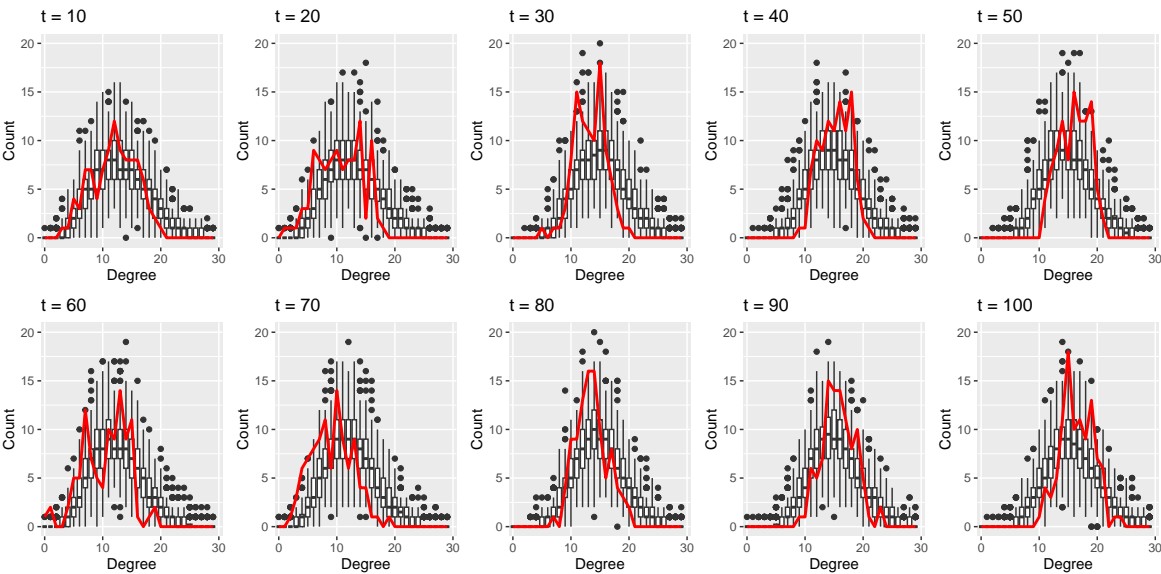

Figure 5: Degree distributions for the generated graphs predicted from decoder at different time points for the number of nodes $n = 100$ and sample size $s = 200$. The red lines correspond to the degree distributions of graphs simulated from STERGM.

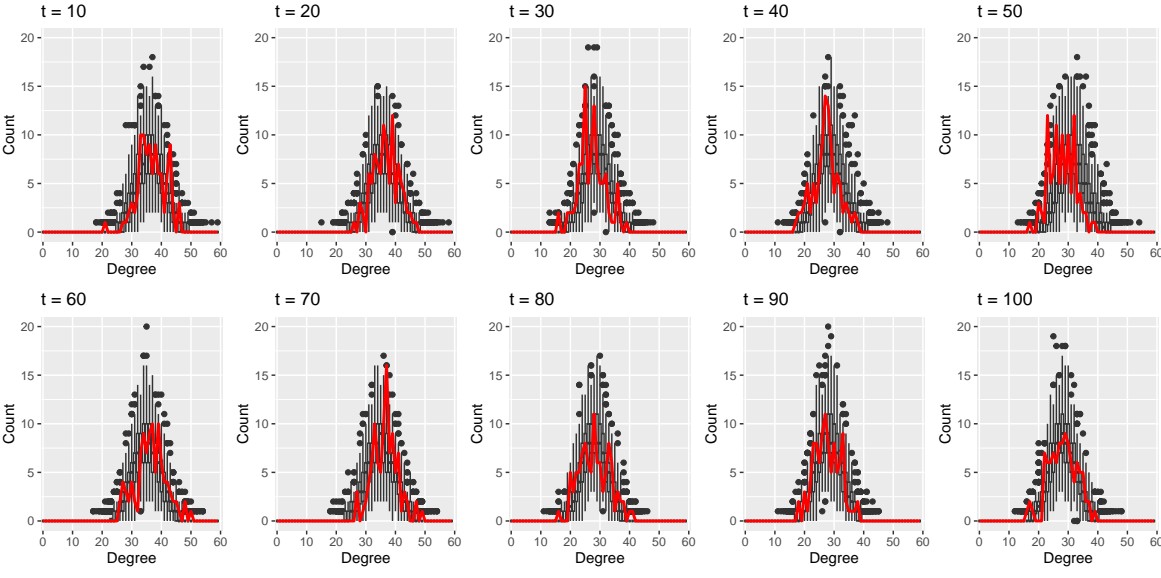

Figure 6: Degree distributions for the generated graphs predicted from decoder at different time points for the number of nodes $n = 100$ and sample size $s = 200$. The red lines correspond to the degree distributions of graphs simulated from SBM.

different detection methods as a supplementary evaluation. Specifically, we fit Degree-Corrected Stochastic Block Models (DCSBM) (Karrer & Newman, 2011; Zhao et al., 2012) to the networks between consecutive detected change points, and we evaluate the log-likelihood of out-of-sample networks that were excluded during model fitting. We choose DCSBM, a generalization of Stochastic Block Model (SBM), because SBM is a well known approximation of graphons, which are among the most general network model in the literature

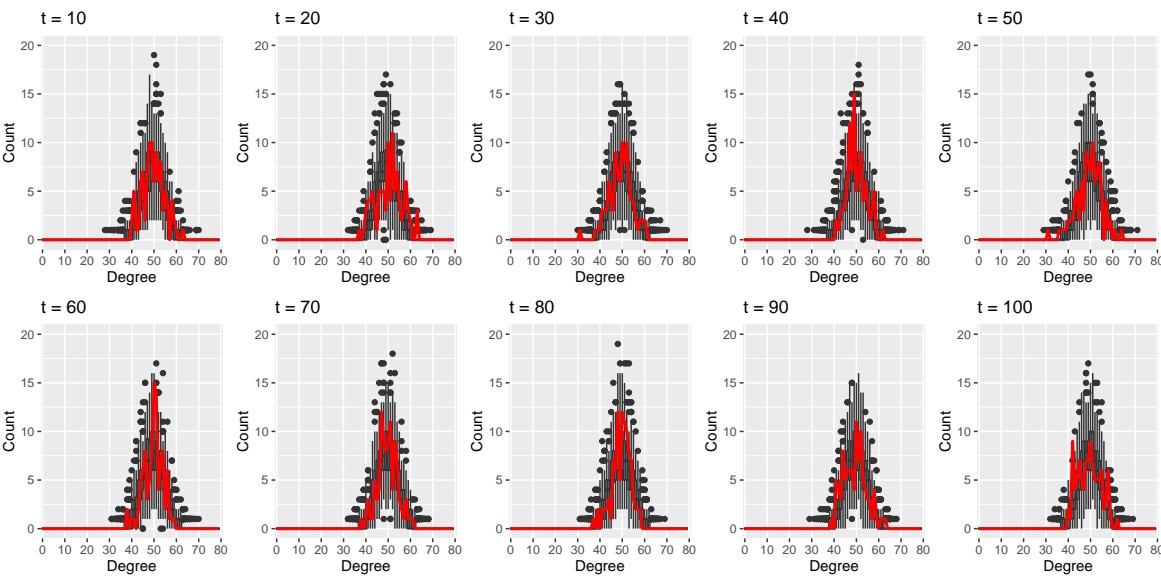

Figure 7: Degree distributions for the generated graphs predicted from decoder at different time points for the number of nodes $n = 100$ and sample size $s = 200$. The red lines correspond to the degree distributions of graphs simulated from RNNs.

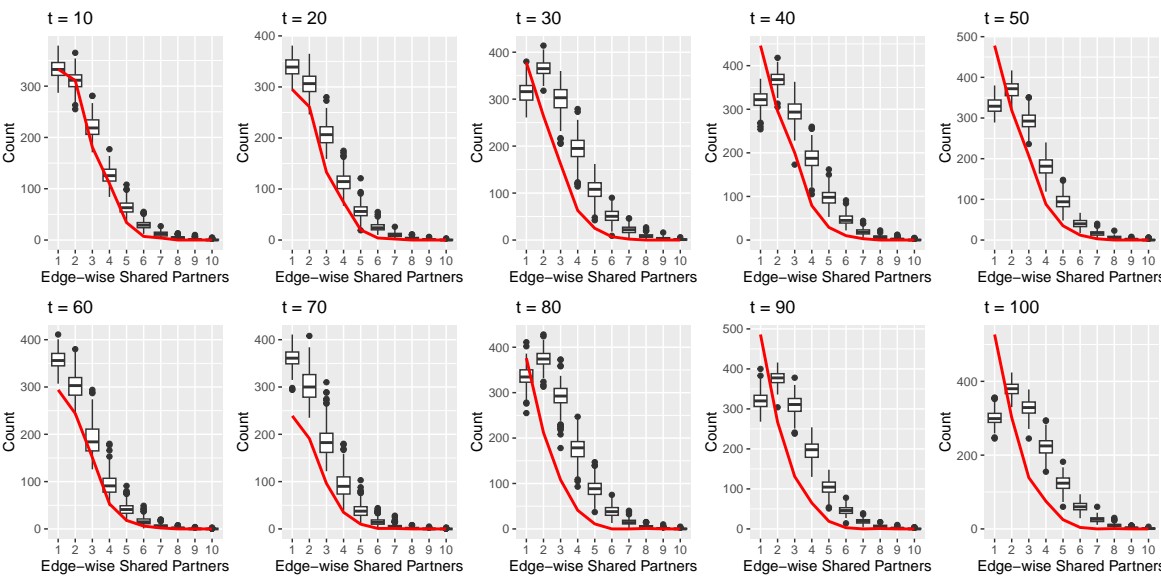

Figure 8: Edge-wise shared partner distributions for the generated graphs predicted from decoder at different time points for the number of nodes $n = 100$ and sample size $s = 200$. The red lines correspond to the edge-wise shared partner distributions of graphs simulated from STERGM.

(Airoldi et al., 2013; Olhede & Wolfe, 2014; Gao et al., 2015). Moreover, DCSBM does not favor either the proposed or competitor methods in terms of fitting the model. Intuitively, a higher log-likelihood suggests that the detected change points segment the time series in a way that better capture the unchanged patterns within each interval. Additional details on this evaluation procedure are provided in Appendix 7.6.

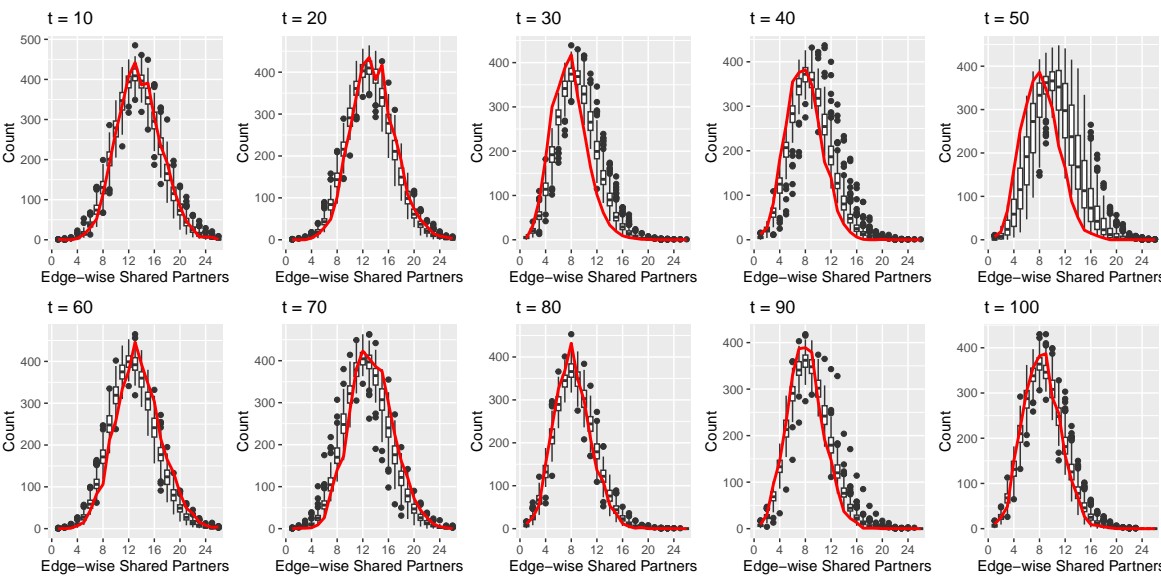

Figure 9: Edge-wise shared partner distributions for the generated graphs predicted from decoder at different time points for the number of nodes $n = 100$ and sample size $s = 200$. The red lines correspond to the edge-wise shared partner distributions of graphs simulated from SBM.

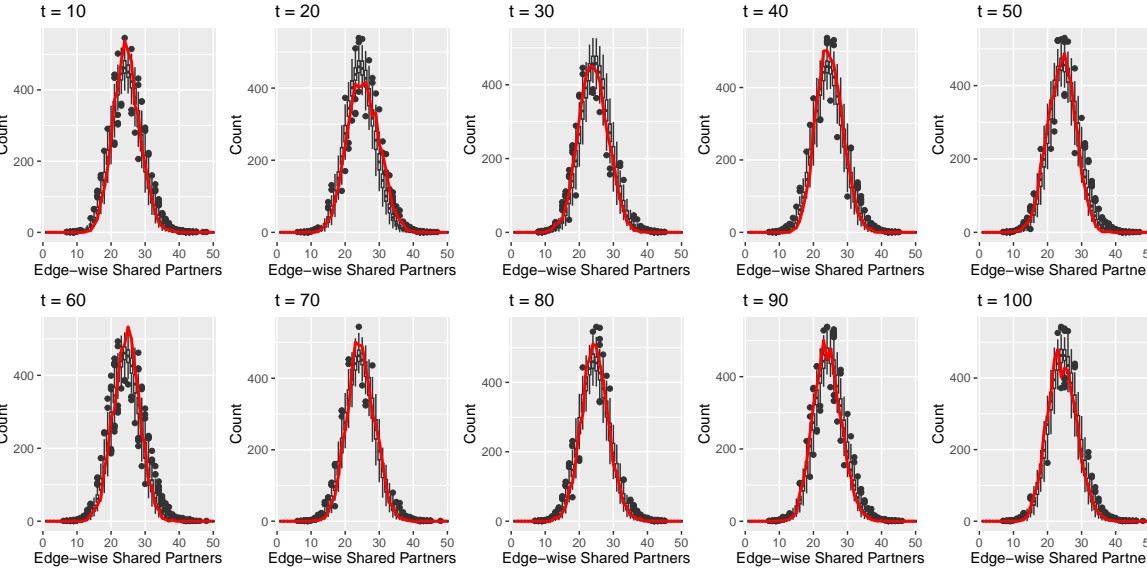

Figure 10: Edge-wise shared partner distributions for the generated graphs predicted from decoder at different time points for the number of nodes $n = 100$ and sample size $s = 200$. The red lines correspond to the edge-wise shared partner distributions of graphs simulated from RNNs.

### 5.3.1 MIT Cellphone Data

The Massachusetts Institute of Technology (MIT) cellphone data (Eagle & Pentland, 2006) depicts human interactions via phone call activities among $n = 96$ participants spanning $T = 232$ days. In the constructed undirected networks, an edge $\boldsymbol{y}_{ij}^t = 1$ indicates that participant $i$ and participant $j$ had made phone calls on

day $t$, and $\boldsymbol{y}_{ij}^t = 0$ otherwise. The data ranges from 2004-09-15 to 2005-05-04, covering the winter break in the MIT academic calendar.

We apply our proposed method to detect change points using the data-driven threshold, and we use network statistics as input data to the gSeg, kerSeg, and CPDstergm methods. Specifically, we use the number of edges, isolates, and triangles to capture the frequency of connections, the sparsity of social interaction, and the transitive association among participants, respectively. Moreover, the CPDrdpg and CPDnbs methods directly utilize networks as input data. Figure 11 displays the magnitude of Equation (14) and the change points detected by the proposed and competitor methods. Table 4 provides a list of potential events, aligning with the detected change points from our method. In general, the magnitude in Figure 11 reflects the scale of structural changes from the observed networks.

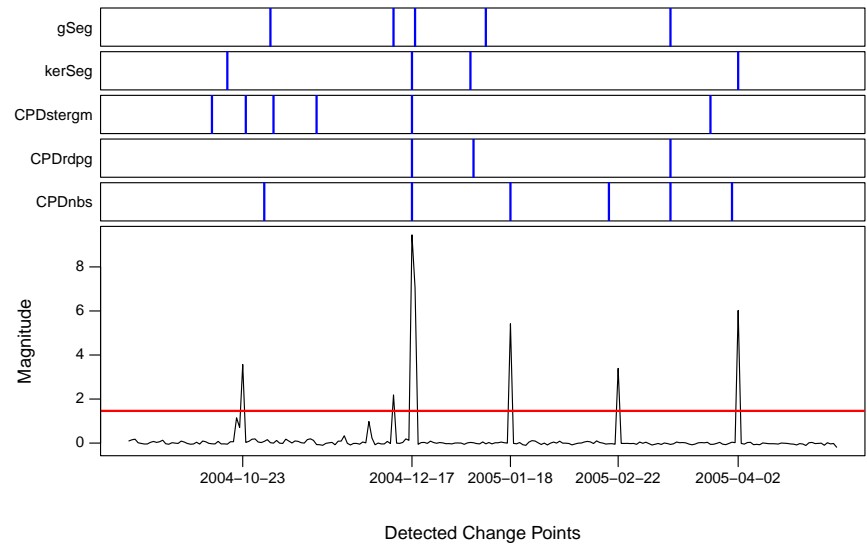

Figure 11: Detected change points from the proposed and competitor (blue) methods on the MIT Cellphone Data. The threshold (red horizontal line) is calculated by (15) with $\mathcal{Z}_{0.9}$. The dates of the detected change points for the competitor methods are displayed in Appendix 7.6.

Without specifying the structural patterns in advance to search for change points, our method can punctually detect the beginning of the winter break, which is the major event that alters the interaction among participants. As the largest spike in our results of Figure 11, the beginning of the winter break is also detected by the competitor methods effectively. Moreover, our method detects a change point on 2004-10-23, corresponding to the annual sponsor meeting that occurred on 2004-10-21. More than two-thirds of the participants have attended the meeting, focusing on achieving project goals throughout the week (Eagle & Pentland, 2006). However, the CPDstergm and CPDrdpg methods struggle to detect this change point. Furthermore, the proposed and competitor methods detect change points related to the spring break, while our method detects two additional change points associated to federal holidays.

The primary discrepancies between the results in Figure 11 are the two federal holidays on 2005-01-17 and 2005-02-21, which are overlooked by some of the competitor methods. In particular, the gSeg, kerSeg, and CPDrdpg methods identify change points slightly earlier than 2005-01-18, while the CPDstergm method does not detect a change point around that period. These discrepancies may be affected by the overlap between the winter break and other holidays around the end of 2004. The detected change points in January 2005 by the competitor methods also suggest that the Martin Luther King day detected by our method is a reasonable change point after the winter break. Moreover, the CPDstergm method detects a clustering of change points during the sponsor meeting around 2004-10-23, which is an indication of overfitting. The result may absorb the excessive noise during that period, such that a clear and single change point cannot be determined. In summary, our result overlaps with the combined results from the competitors, validating the effectiveness of the proposed methods.

Table 4: Potential nearby events aligned with the detected change points from our proposed method on the MIT cellphone data.

| Detected change points | Potential nearby events |
|---|---|
| 2004-10-23 | 2004-10-21 Sponsor meeting |
| 2004-12-17 | 2004-12-18 to 2005-01-02 Winter break |
| 2005-01-18 | 2005-01-17 Martin Luther King Day |
| 2005-02-22 | 2005-02-21 Presidents Day |
| 2005-04-02 | 2005-03-21 to 2005-03-25 Spring break |

Finally, Table 5 presents a comparison of the log-likelihood values for out-of-sample graphs evaluated using the fitted DCSBM. Although this evaluation procedure is heuristic, the log-likelihood can potentially indicate how well the detected change points capture the unchanged patterns within each segmented interval. To assess robustness and sensitivity, we also compare the results by selectively removing graphs at different time gaps, specifically $\Delta t = \{15, 20, 25, 30\}$ for $T = 232$. The higher log-likelihood values associated with change points detected by our method suggest that it identifies more meaningful segmentation compared to competitor methods.

Table 5: Log-likelihood values of the out-of-sample graphs evaluated using the fitted DCSBM corresponding to their respective intervals in the MIT cellphone data.

| $\Delta t$ | CPDlatent | CPDnbs | CPDrdpg | CPDstergm | kerSeg | gSeg |
|---|---|---|---|---|---|---|
| 15 | $-3593.38$ | $-3595.74$ | $-3968.61$ | $-3718.24$ | $-4026.17$ | $-3891.61$ |
| 20 | $-2441.86$ | $-2617.12$ | $-2612.46$ | $-2760.73$ | $-2592.96$ | $-2604.36$ |
| 25 | $-1903.14$ | $-2056.03$ | $-2067.39$ | $-2172.59$ | $-2005.87$ | $-2123.85$ |
| 30 | $-1869.47$ | $-1901.00$ | $-1956.27$ | $-1906.28$ | $-1991.85$ | $-1914.86$ |

### 5.3.2 Enron Email Data

The Enron email data, analyzed by Priebe et al. (2005), Park et al. (2012), and Peel & Clauset (2015), portrays the communication patterns among employees before the collapse of a giant energy company. The data of our focus consists of $T = 100$ weekly undirected networks, ranging from 2000-06-05 to 2002-05-06 for $n = 100$ employees. We use the same configuration as described in Section 5.3.1 for the proposed and competitor methods to detect change points. Figure 12 displays the magnitude of Equation (14) and the detected change points from the proposed and competitor methods. Furthermore, Table 6 provides a list of potential events, aligning with the detected change points from our method. In general, the magnitude in Figure 12 reflects the scale of structural changes from the observed networks.

Table 6: Potential nearby events aligned with the detected change points from our proposed method on the Enron email data.

| Detected change points | Potential nearby events |
|---|---|
| 2000-10-16 | 2000-11-01 FERC exonerated Enron |
| 2001-06-11 | 2001-06-21 CEO publicly confronted |
| 2001-09-24 | 2001-09-26 Internal employee meeting |
| 2001-12-03 | 2001-12-02 Enron filed for bankruptcy |

In 2001, Enron underwent a multitude of major and overlapping incidents, making it difficult to associate the detected change points with specific real events. Despite the turmoil, our method detects four signifi-

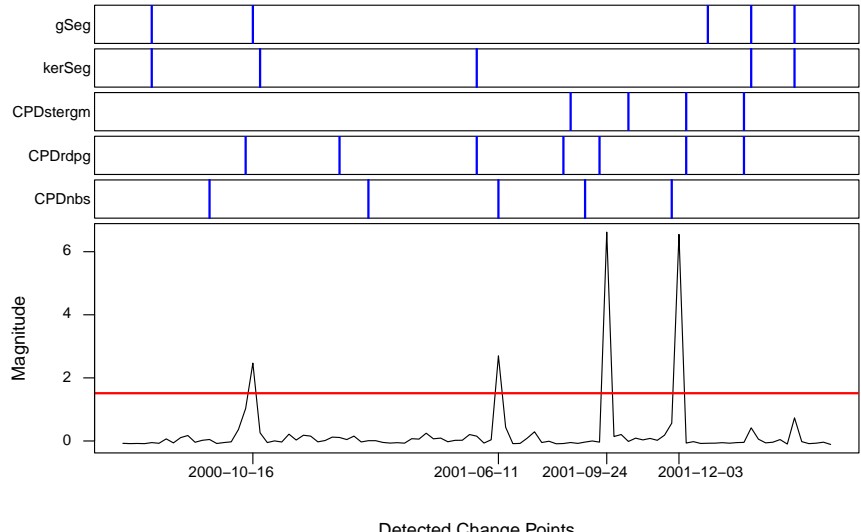

Figure 12: Detected change points from the proposed and competitor (blue) methods on the Enron email data. The threshold (red horizontal line) is calculated by (15) with $\mathcal{Z}_{0.9}$. The dates of the detected change points for the competitor methods are displayed in Appendix 7.6.

cant change points that closely align with pivotal moments in Enron's timeline. Throughout 2000, Enron orchestrated rolling blackouts, causing staggering surges in electricity prices that peaked at twenty times the standard rate. Thence, the first change point, detected on 2000-10-16 by our method, aligns with the Federal Energy Regulatory Commission (FERC) exonerating Enron of wrongdoing on 2000-11-01. As a major event with chain reaction throughout 2000, this change point is also detected by the gSeg, kerSeg, CPDrdpg, and CPDnbs methods. Subsequently, a second change point, detected on 2001-06-11, aligns with the CEO confronted by an activist on 2001-06-21 in protesting against Enron's role in the energy crisis. This public incident is also detected by the kerSeg, CPDrdpg, and CPDnbs methods while overlooked by the other competitor methods.

The next two change points are associated with more pronounced shifts in network patterns, indicated by the two substantially large spikes in Figure 12. Specifically, the third change point, detected on 2001-09-24, coincides with an internal employee meeting on 2001-09-26, during which the CEO reassured employees that Enron's stock was a good buy and the company's accounting methods were legal and appropriate. Following this meeting, Enron's stock saw a final surge before continuing its sharp decline. Finally, our method detects a change point on 2001-12-03, aligning with Enron filing for bankruptcy on 2001-12-02, marking the collapse of the largest energy company in the U.S.

Based on the results in Figure 12, the primary discrepancies between the proposed and competitor methods are observed at the endpoint of the time span. In particular, four competitor methods have detected change points in February 2002, corresponding to events after Enron filed for bankruptcy in December 2001. While the magnitude exhibits two small spikes on the right in Figure 12, they do not exceed the threshold in red to be declared as change points by our method. The scale of changes in communication patterns among employees is smaller for the events in February 2002, comparing to the scale of changes in communication patterns during bankruptcy.

Besides lowering the threshold to include these change points, the discrepancy suggests a potential extension to improve the proposed framework. Perhaps we could specify the priors for the graph-level representations as $\boldsymbol{z}^t \sim \mathcal{N}(\boldsymbol{\mu}^t, \boldsymbol{\Sigma}^t)$ and incorporate both the mean $\boldsymbol{\mu}^t$ and covariance $\boldsymbol{\Sigma}^t$ into the penalty term of the objective function. For example, to adaptively adjust the changes at different scales, the inverse of the covariance $\boldsymbol{\Sigma}^t \in \mathbb{R}^{d \times d}$ may serve as a scaling factor for the difference $\boldsymbol{\mu}^{t+1} - \boldsymbol{\mu}^t \in \mathbb{R}^d$, thereby enhancing the magnitudes. By learning the covariances of the graph-level representations and monitoring their consecutive

shifts over time, the latent space regularized in this way could potentially capture more subtle variations in the data space. Furthermore, the ADMM procedure would require an update for $\boldsymbol{\Sigma}^t$, and the quadratic form for the localization method would need to be modified as $(\boldsymbol{z}^t - \boldsymbol{z}^{t-1})^\top (\boldsymbol{\Sigma}^t + \boldsymbol{\Sigma}^{t-1})^{-1}(\boldsymbol{z}^t - \boldsymbol{z}^{t-1}) \sim \chi_d^2$ where the inverse of the covariance is also used for standardization. As there are different ways to incorporate the covariances into the regularization term and learning the positive semi-definite covariance via neural network is challenging, we consider this improvement as a potential direction for future development.

The result for gSeg and kerSeg methods can be affected by the choices of network statistics, which may not be representative enough to capture the structural changes. For CPDstergm, the absence of change points in 2000 and the clustering of four change points in late 2001 indicate the detection is sensitive to the noise where network structures shift rapidly during bankruptcy, rendering the detected change points unreliable as the model is overfitted. In summary, the overlap between the proposed and competitor methods validates the effectiveness of the proposed methods.

Finally, Table 7 compares the log-likelihood with the fitted DCSBM across different methods. Although this is a heuristic evaluation approach, the log-likelihood values provide information into how well the detected change points maintain structural coherence within each interval. To assess robustness and sensitivity, we examine the results by selectively removing graphs at varying time gaps, specifically $\Delta t = \{3, 6, 9, 12\}$ for $T = 100$. The log-likelihood values based on the change points detected by our method are higher most of the time, suggesting that our approach identifies more reasonable change points compared to competitor methods heuristically.

Table 7: Log-likelihood values of the out-of-sample graphs evaluated using the fitted DCSBM corresponding to their respective intervals in the Enron email data.

| $\Delta t$ | CPDlatent | CPDnbs | CPDrdgp | CPDstergm | kerSeg | gSeg |
|---|---|---|---|---|---|---|
| 3 | $-9863.13$ | $-10340.97$ | $-10485.42$ | $-11681.16$ | $-9936.28$ | $-11697.72$ |
| 6 | $-5068.16$ | $-4944.41$ | $-4821.88$ | $-5558.92$ | $-5025.10$ | $-4967.78$ |
| 9 | $-2994.83$ | $-3315.66$ | $-3238.66$ | $-3190.70$ | $-3217.85$ | $-3274.96$ |
| 12 | $-2482.73$ | $-2507.28$ | $-2570.34$ | $-2518.89$ | $-2540.31$ | $-2592.20$ |

## 6 Discussion

This paper proposes to detect change points in time series of graphs using a decoder-only latent space model. Intrinsically, dynamic network structures can be complex due to dyadic and temporal dependencies, making inference for dynamic graphs a challenging task. Learning low-dimensional graph representations can extract useful features to facilitate change point detection in time series of graphs. Specifically, we assume each observed network is generated from a latent variable through a graph decoder. We also impose prior distributions to the graph-level representations, and the priors are learned from the data as empirical Bayes. The optimization problem with Group Fused Lasso penalty on the prior parameters is solved via ADMM, and experiment results demonstrate that generative model is useful for change point detection.

Several extensions to our proposed framework are possible for future development. Besides binary networks, relations by nature have degree of strength, which are denoted by generic values. Moreover, nodal and dyadic attributes are important components in network data. Hence, models that can generate weighted edges, as well as nodal and dyadic attributes, can capture more information about the network dynamics (Fellows & Handcock, 2012; Krivitsky, 2012; Kei et al., 2023a). Furthermore, the number of nodes and their attributes are subjected to change over time. Extending the framework to allow varying network size and to detect node-level anomalies can provide granular insights of network changes (Simonovsky & Komodakis, 2018; Shen et al., 2023). Also, improving the scalability and computational efficiency for representation learning is crucial (Killick et al., 2012; Gallagher et al., 2021), especially for handling large and weighted graphs. While our framework demonstrates the ability in change point detection, the development of more sophisticated

neural network architectures can enhance the model's capacity on other meaningful tasks (Handcock et al., 2007; Kolar et al., 2010; Yu et al., 2021; Madrid Padilla et al., 2023).

## Code Availability

The codes are available at `https://github.com/allenkei/CPD_generative`.

## Acknowledgments

We thank Mark Handcock and Ying Nian Wu for helpful comments on this work. Also, we are extremely grateful for the constructive comments provided by the anonymous reviewers and Action Editors.

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

# 7 Appendix

## 7.1 Updating $\mu$ and $\phi$

In this section, we derive the updates for prior parameter $\boldsymbol{\mu} \in \mathbb{R}^{T \times d}$ and graph decoder parameter $\boldsymbol{\phi}$. Denote the objective function in Equation (6) as $\mathcal{L}(\boldsymbol{\phi}, \boldsymbol{\mu})$ and denote the set of parameters $\{\boldsymbol{\phi}, \boldsymbol{\mu}\}$ as $\boldsymbol{\theta}$. We first calculate the gradient of log-likelihood $l(\boldsymbol{\theta})$ in $\mathcal{L}(\boldsymbol{\phi}, \boldsymbol{\mu})$ with respect to $\boldsymbol{\theta}$:

$$
\begin{aligned}
\nabla_{\boldsymbol{\theta}} \, l(\boldsymbol{\theta}) &= \nabla_{\boldsymbol{\theta}} \sum_{t=1}^{T} \log P(\boldsymbol{y}^t) \\
&= \sum_{t=1}^{T} \frac{1}{P(\boldsymbol{y}^t)} \nabla_{\boldsymbol{\theta}} P(\boldsymbol{y}^t) \\
&= \sum_{t=1}^{T} \frac{1}{P(\boldsymbol{y}^t)} \nabla_{\boldsymbol{\theta}} \int P(\boldsymbol{y}^t, \boldsymbol{z}^t) d\boldsymbol{z}^t \\
&= \sum_{t=1}^{T} \frac{1}{P(\boldsymbol{y}^t)} \int P(\boldsymbol{y}^t, \boldsymbol{z}^t) \Big[ \nabla_{\boldsymbol{\theta}} \log P(\boldsymbol{y}^t, \boldsymbol{z}^t) \Big] d\boldsymbol{z}^t \\
&= \sum_{t=1}^{T} \int \frac{P(\boldsymbol{y}^t, \boldsymbol{z}^t)}{P(\boldsymbol{y}^t)} \Big[ \nabla_{\boldsymbol{\theta}} \log P(\boldsymbol{y}^t, \boldsymbol{z}^t) \Big] d\boldsymbol{z}^t \\
&= \sum_{t=1}^{T} \int P(\boldsymbol{z}^t | \boldsymbol{y}^t) \Big[ \nabla_{\boldsymbol{\theta}} \log P(\boldsymbol{y}^t, \boldsymbol{z}^t) \Big] d\boldsymbol{z}^t \\
&= \sum_{t=1}^{T} \mathbb{E}_{P(\boldsymbol{z}^t | \boldsymbol{y}^t)} \Big( \nabla_{\boldsymbol{\theta}} \log \Big[ P(\boldsymbol{y}^t | \boldsymbol{z}^t) P(\boldsymbol{z}^t) \Big] \Big) \\
&= \sum_{t=1}^{T} \mathbb{E}_{P(\boldsymbol{z}^t | \boldsymbol{y}^t)} \Big( \nabla_{\boldsymbol{\theta}} \log P(\boldsymbol{y}^t | \boldsymbol{z}^t) \Big) + \sum_{t=1}^{T} \mathbb{E}_{P(\boldsymbol{z}^t | \boldsymbol{y}^t)} \Big( \nabla_{\boldsymbol{\theta}} \log P(\boldsymbol{z}^t) \Big).
\end{aligned}
$$

Note that the expectation in the gradient is now with respect to the posterior distribution $P(\boldsymbol{z}^t | \boldsymbol{y}^t) \propto P(\boldsymbol{y}^t | \boldsymbol{z}^t) \times P(\boldsymbol{z}^t)$. Furthermore, the gradient of $\mathcal{L}(\boldsymbol{\phi}, \boldsymbol{\mu})$ with respect to the prior parameter $\boldsymbol{\mu}^t \in \mathbb{R}^d$ at a specific time point $t$ is

$$
\begin{aligned}
\nabla_{\boldsymbol{\mu}^t} \, \mathcal{L}(\boldsymbol{\phi}, \boldsymbol{\mu}) &= -\mathbb{E}_{P(\boldsymbol{z}^t | \boldsymbol{y}^t)} \Big( \nabla_{\boldsymbol{\mu}^t} \log P(\boldsymbol{z}^t) \Big) + \kappa(\boldsymbol{\mu}^t - \boldsymbol{\nu}^t + \boldsymbol{w}^t) \\
&= -\mathbb{E}_{P(\boldsymbol{z}^t | \boldsymbol{y}^t)} (\boldsymbol{z}^t - \boldsymbol{\mu}^t) + \kappa(\boldsymbol{\mu}^t - \boldsymbol{\nu}^t + \boldsymbol{w}^t).
\end{aligned}
$$

Setting the gradient $\nabla_{\boldsymbol{\mu}^t} \, \mathcal{L}(\boldsymbol{\phi}, \boldsymbol{\mu})$ to zeros and solve for $\boldsymbol{\mu}^t$, we have

$$
\begin{aligned}
\boldsymbol{0} &= -\mathbb{E}_{P(\boldsymbol{z}^t | \boldsymbol{y}^t)} (\boldsymbol{z}^t) + (1 + \kappa) \boldsymbol{\mu}^t - \kappa(\boldsymbol{\nu}^t - \boldsymbol{w}^t) \\
(1 + \kappa) \boldsymbol{\mu}^t &= \mathbb{E}_{P(\boldsymbol{z}^t | \boldsymbol{y}^t)} (\boldsymbol{z}^t) + \kappa(\boldsymbol{\nu}^t - \boldsymbol{w}^t) \\
\boldsymbol{\mu}^t &= \frac{1}{1 + \kappa} \mathbb{E}_{P(\boldsymbol{z}^t | \boldsymbol{y}^t)} (\boldsymbol{z}^t) + \frac{\kappa}{1 + \kappa} (\boldsymbol{\nu}^t - \boldsymbol{w}^t).
\end{aligned}
$$

Evidently, the gradient of $\mathcal{L}(\boldsymbol{\phi}, \boldsymbol{\mu})$ with respect to the graph decoder parameter $\boldsymbol{\phi}$ is

$$\nabla_{\boldsymbol{\phi}} \, \mathcal{L}(\boldsymbol{\phi}, \boldsymbol{\mu}) = -\sum_{t=1}^{T} \mathbb{E}_{P(\boldsymbol{z}^t|\boldsymbol{y}^t)} \Big( \nabla_{\boldsymbol{\phi}} \log P(\boldsymbol{y}^t|\boldsymbol{z}^t) \Big).$$

The parameter $\boldsymbol{\phi}$ can be updated efficiently through back-propagation.

## 7.2  Langevin Dynamics

Calculating the solution in (9) and the gradient in (10) requires evaluating the conditional expectations under the posterior distribution $P(\boldsymbol{z}^t|\boldsymbol{y}^t) \propto P(\boldsymbol{y}^t|\boldsymbol{z}^t) \times P(\boldsymbol{z}^t)$. In this section, we discuss the Langevin Dynamics to sample $\boldsymbol{z}^t \in \mathbb{R}^d$ from the posterior distribution $P(\boldsymbol{z}^t|\boldsymbol{y}^t)$ that is conditional on the observed network $\boldsymbol{y}^t \in \{0,1\}^{n \times n}$. The Langevin Dynamics, a short run MCMC, is achieved by iterating the following:

$$\begin{aligned}
\boldsymbol{z}_{\tau+1}^t &= \boldsymbol{z}_{\tau}^t + \delta \big[ \nabla_{\boldsymbol{z}^t} \log P(\boldsymbol{z}^t|\boldsymbol{y}^t) \big] + \sqrt{2\delta}\boldsymbol{\epsilon} \\
&= \boldsymbol{z}_{\tau}^t + \delta \big[ \nabla_{\boldsymbol{z}^t} \log P(\boldsymbol{y}^t|\boldsymbol{z}^t) + \nabla_{\boldsymbol{z}^t} \log P(\boldsymbol{z}^t) - \nabla_{\boldsymbol{z}^t} \log P(\boldsymbol{y}^t) \big] + \sqrt{2\delta}\boldsymbol{\epsilon} \\
&= \boldsymbol{z}_{\tau}^t + \delta \big[ \nabla_{\boldsymbol{z}^t} \log P(\boldsymbol{y}^t|\boldsymbol{z}^t) - (\boldsymbol{z}_{\tau}^t - \boldsymbol{\mu}^t) \big] + \sqrt{2\delta}\boldsymbol{\epsilon}
\end{aligned}$$

where $\tau$ is the time step and $\delta$ is the step size of the Langevin Dynamics. The error term $\boldsymbol{\epsilon} \sim \mathcal{N}(\boldsymbol{0}, \boldsymbol{I}_d)$ serves as a random perturbation to the sampling process. The gradient of the graph decoder $P(\boldsymbol{y}^t|\boldsymbol{z}^t)$ with respect to the latent variable $\boldsymbol{z}^t$ can be calculated efficiently through back-propagation. Essentially, we use MCMC samples to approximate the conditional expectation $\mathbb{E}_{P(\boldsymbol{z}^t|\boldsymbol{y}^t)}(\cdot)$ in the solution (9) and the gradient (10).

## 7.3  Group Lasso for Updating $\beta$

In this section, we present the derivation to update $\boldsymbol{\beta}$ in Proposition 2, which is equivalent to solving a Group Lasso problem (Yuan & Lin, 2006). We adapt the derivation from Bleakley & Vert (2011) for our proposed ADMM algorithm. Denote the objective function in (7) as $\mathcal{L}(\boldsymbol{\gamma}, \boldsymbol{\beta})$. When $\boldsymbol{\beta}_{t,\cdot} \neq \boldsymbol{0}$, the gradient of $\mathcal{L}(\boldsymbol{\gamma}, \boldsymbol{\beta})$ with respect to $\boldsymbol{\beta}_{t,\cdot}$ is

$$\nabla_{\boldsymbol{\beta}_{t,\cdot}} \mathcal{L}(\boldsymbol{\gamma}, \boldsymbol{\beta}) = \lambda \frac{\boldsymbol{\beta}_{t,\cdot}}{\|\boldsymbol{\beta}_{t,\cdot}\|_2} - \kappa \boldsymbol{X}_{\cdot,t}^{\top}(\boldsymbol{\mu}_{(a+1)} + \boldsymbol{w}_{(a)} - \boldsymbol{1}_{T,1}\boldsymbol{\gamma} - \boldsymbol{X}_{\cdot,t}\boldsymbol{\beta}_{t,\cdot} - \boldsymbol{X}_{\cdot,-t}\boldsymbol{\beta}_{-t,\cdot})$$

where $\boldsymbol{X}_{\cdot,t} \in \mathbb{R}^{T \times 1}$ is the $t$-th column of matrix $\boldsymbol{X} \in \mathbb{R}^{T \times (T-1)}$ and $\boldsymbol{\beta}_{t,\cdot} \in \mathbb{R}^{1 \times d}$ is the $t$-th row of matrix $\boldsymbol{\beta} \in \mathbb{R}^{(T-1) \times d}$. Moreover, we denote $\boldsymbol{\beta}_{-t,\cdot} \in \mathbb{R}^{(T-1) \times p}$ as the matrix obtained by replacing the $t$-th row of matrix $\boldsymbol{\beta}$ with a zero vector, and $\boldsymbol{X}_{\cdot,-t} \in \mathbb{R}^{T \times (T-1)}$ is denoted similarly.

Setting the above gradient to zeros, we have

$$\boldsymbol{\beta}_{t,\cdot} = \Big( \kappa \boldsymbol{X}_{\cdot,t}^{\top}\boldsymbol{X}_{\cdot,t} + \frac{\lambda}{\|\boldsymbol{\beta}_{t,\cdot}\|_2} \Big)^{-1} \boldsymbol{b}_t \tag{16}$$

where

$$\boldsymbol{b}_t = \kappa \boldsymbol{X}_{\cdot,t}^{\top}(\boldsymbol{\mu}_{(a+1)} + \boldsymbol{w}_{(a)} - \boldsymbol{1}_{T,1}\boldsymbol{\gamma} - \boldsymbol{X}_{\cdot,-t}\boldsymbol{\beta}_{-t,\cdot}) \in \mathbb{R}^{1 \times d}.$$

Calculating the Euclidean norm of (16) on both sides and rearrange the terms, we have

$$\|\boldsymbol{\beta}_{t,\cdot}\|_2 = (\kappa \boldsymbol{X}_{\cdot,t}^{\top}\boldsymbol{X}_{\cdot,t})^{-1}(\|\boldsymbol{b}_t\|_2 - \lambda).$$

Plugging $\|\boldsymbol{\beta}_{t,\cdot}\|_2$ into (16) for substitution, the solution of $\boldsymbol{\beta}_{t,\cdot}$ is arrived at

$$\boldsymbol{\beta}_{t,\cdot} = \frac{1}{\kappa \boldsymbol{X}_{\cdot,t}^{\top}\boldsymbol{X}_{\cdot,t}} \Big( 1 - \frac{\lambda}{\|\boldsymbol{b}_t\|_2} \Big) \boldsymbol{b}_t.$$

Moreover, when $\boldsymbol{\beta}_{t,\cdot} = \boldsymbol{0}$, the subgradient $\boldsymbol{v}$ of $\|\boldsymbol{\beta}_{t,\cdot}\|_2$ needs to satisfy that $\|\boldsymbol{v}\|_2 \leq 1$. Because

$$\boldsymbol{0} \in \lambda\boldsymbol{v} - \kappa \boldsymbol{X}_{\cdot,t}^{\top}(\boldsymbol{\mu}_{(a+1)} + \boldsymbol{w}_{(a)} - \boldsymbol{1}_{T,1}\boldsymbol{\gamma} - \boldsymbol{X}_{\cdot,-t}\boldsymbol{\beta}_{-t,\cdot}),$$

we obtain the condition that $\boldsymbol{\beta}_{t,\cdot}$ becomes $\mathbf{0}$ when $\|\boldsymbol{b}_t\|_2 \leq \lambda$. Therefore, we can iteratively apply the following to update $\boldsymbol{\beta}_{t,\cdot}$ for each block $t = 1, \ldots, T-1$:

$$\boldsymbol{\beta}_{t,\cdot} \leftarrow \frac{1}{\kappa \boldsymbol{X}_{\cdot,t}^\top \boldsymbol{X}_{\cdot,t}} \left(1 - \frac{\lambda}{\|\boldsymbol{b}_t\|_2}\right)_+ \boldsymbol{b}_t$$

where $(\cdot)_+ = \max(\cdot, 0)$.

### 7.4 ADMM Procedure

The procedure to solve the problem in (3) via ADMM is presented in Algorithm 1. The steps to transform between $\boldsymbol{\nu}$ and $(\boldsymbol{\gamma}, \boldsymbol{\beta})$ within an ADMM iteration are omitted for succinctness. The complexity of the proposed algorithm is at least of order $O\big(A(Tsld + BTnk + DT)\big)$ with additional gradient calculation for neural networks in sub-routines. Specifically, for each of $A$ iterations of ADMM, we update the prior parameter $\boldsymbol{\mu}^t \in \mathbb{R}^d$ for all $T$ time points, and each update involves $l$ steps of MCMC for $s$ samples. Then we calculate the gradients of neural networks for all $T$ time points and run $B$ iterations of Adam optimizer. The output of neural networks has dimensions $n$ by $k$. Lastly, we run $D$ iterations of block coordinate descent for the sequential differences. Essentially, ADMM decomposes a complex optimization problem into smaller problems, targeting individual component one at a time.

---

**Algorithm 1** Latent Space Group Fused Lasso

---

1: **Input**: learning iterations $A, B, D$, tuning parameter $\lambda$, penalty parameter $\kappa$, learning rates $\eta$, observed data $\{\boldsymbol{y}^t\}_{t=1}^T$, initialization $\{\boldsymbol{\phi}_{(1)}, \boldsymbol{\mu}_{(1)}, \boldsymbol{\gamma}_{(1)}, \boldsymbol{\beta}_{(1)}, \boldsymbol{w}_{(1)}\}$
2: **for** $a = 1, \cdots, A$ **do**
3:     **for** $t = 1, \cdots, T$ **do**
4:         draw $s$ samples $\boldsymbol{z}_1^t, \ldots, \boldsymbol{z}_s^t$ from $P(\boldsymbol{z}^t | \boldsymbol{y}^t)$ according to (11)
5:         $\boldsymbol{\mu}_{(a+1)}^t = \frac{1}{1+\kappa}(s^{-1}\sum_{i=1}^s \boldsymbol{z}_i^t) + \frac{\kappa}{1+\kappa}(\boldsymbol{\nu}^t - \boldsymbol{w}^t)$
6:     **end for**
7:     **for** $b = 1, \ldots, B$ **do**
8:         $\boldsymbol{\phi}_{(b+1)} = \boldsymbol{\phi}_{(b)} - \eta \times \nabla_{\boldsymbol{\phi}} \mathcal{L}(\boldsymbol{\phi}, \boldsymbol{\mu})$
9:     **end for**
10:    Set $\tilde{\boldsymbol{\gamma}}^{(1)} = \boldsymbol{\gamma}_{(a)}$ and $\tilde{\boldsymbol{\beta}}^{(1)} = \boldsymbol{\beta}_{(a)}$
11:    **for** $d = 1, \ldots, D$ **do**
12:       Let $\tilde{\boldsymbol{\beta}}_{t,\cdot}^{(d+1)}$ be updated according to (12) for $t = 1, \ldots, T-1$
13:       $\tilde{\boldsymbol{\gamma}}^{(d+1)} = (1/T)\mathbf{1}_{1,T} \cdot (\boldsymbol{\mu}_{(a+1)} + \boldsymbol{w}_{(a)} - \boldsymbol{X}\tilde{\boldsymbol{\beta}}^{(d+1)})$
14:    **end for**
15:    Set $\boldsymbol{\gamma}_{(a+1)} = \tilde{\boldsymbol{\gamma}}^{(d+1)}$ and $\boldsymbol{\beta}_{(a+1)} = \tilde{\boldsymbol{\beta}}^{(d+1)}$
16:    $\boldsymbol{w}_{(a+1)} = \boldsymbol{\mu}_{(a+1)} - \boldsymbol{\nu}_{(a+1)} + \boldsymbol{w}_{(a)}$
17: **end for**
18: $\hat{\boldsymbol{\mu}} \leftarrow \boldsymbol{\mu}_{(a+1)}$
19: **Output**: learned prior parameters $\hat{\boldsymbol{\mu}}$

---

### 7.5 Practical Guidelines

#### 7.5.1 ADMM Implementation

In this section, we provide practical guidelines for the proposed framework and ADMM algorithm. For Langevin Dynamic sampling, we set $\delta = 0.5$, and we draw $s = 200$ samples for each time point $t$. To detect change points using the data-driven threshold in (15), we let the tuning parameter $\lambda = \{10, 20, 50, 100\}$. To detect change points using the localizing method with Gamma distribution in (13), we let the tuning parameter $\lambda = \{5, 10, 20, 50\}$. For each $\lambda$, we run $A = 50$ iterations of ADMM. Within each ADMM iteration, we run $B = 20$ iterations of gradient descent with Adam optimizer for the graph decoder and $D = 20$ iterations of block coordinate descent for Group Lasso. We run our experiment with Tesla T4 GPU. The running time for the simulated study is about two hours for a scenario with all sequences and

cross-validation on the tuning parameter $\lambda$. The running time for the real data experiment is approximately twenty to thirty minutes including cross-validation on the tuning parameter $\lambda$.

Since the proposed generative model is a probability distribution for the observed network data, in this work we stop ADMM learning with the following stopping criteria:

$$\left| \frac{l(\boldsymbol{\phi}_{(a+1)}, \boldsymbol{\mu}_{(a+1)}) - l(\boldsymbol{\phi}_{(a)}, \boldsymbol{\mu}_{(a)})}{l(\boldsymbol{\phi}_{(a)}, \boldsymbol{\mu}_{(a)})} \right| \le \epsilon_{\text{tol}}. \tag{17}$$

The log-likelihood $l(\boldsymbol{\phi}, \boldsymbol{\mu})$ is approximated by sampling from the prior distribution $p(\boldsymbol{z}^t)$, as described in Section 4.2. Hence, we stop the ADMM procedure until the above criteria is satisfied for $a'$ consecutive iterations. In Section 5, we set $\epsilon_{\text{tol}} = 10^{-5}$ and $a' = 5$.

Here we also elaborate on the computational aspect of the approximation of the log-likelihood. To calculate the product of edge probabilities for the conditional distribution $P(\boldsymbol{y}^t|\boldsymbol{z}^t)$, we have the following:

$$
\begin{aligned}
\sum_{t=1}^{T} \log P(\boldsymbol{y}^t) &= \sum_{t=1}^{T} \log \int P(\boldsymbol{y}^t|\boldsymbol{z}^t)P(\boldsymbol{z}^t)d\boldsymbol{z}^t \\
&= \sum_{t=1}^{T} \log \mathbb{E}_{P(\boldsymbol{z}^t)}\big[ \prod_{(i,j)\in\mathbb{Y}} P(\boldsymbol{y}_{ij}^t|\boldsymbol{z}^t)\big] \\
&\approx \sum_{t=1}^{T} \log \Big[\frac{1}{s}\sum_{u=1}^{s}\big[ \prod_{(i,j)\in\mathbb{Y}} P(\boldsymbol{y}_{ij}^t|\boldsymbol{z}_u^t)\big]\Big] \\
&= \sum_{t=1}^{T} \log \Big[\frac{1}{s}\sum_{u=1}^{s}\exp\{ \sum_{(i,j)\in\mathbb{Y}} \log[P(\boldsymbol{y}_{ij}^t|\boldsymbol{z}_u^t)]\}\Big] \\
&= \sum_{t=1}^{T}\Big\{ -\log s + \log\Big[ \exp C^t \sum_{u=1}^{s}\exp\{ \sum_{(i,j)\in\mathbb{Y}} \log[P(\boldsymbol{y}_{ij}^t|\boldsymbol{z}_u^t)] - C^t\}\Big]\Big\} \\
&= \sum_{t=1}^{T}\Big\{ C^t + \log\Big[ \sum_{u=1}^{s}\exp\{ \sum_{(i,j)\in\mathbb{Y}} \log[P(\boldsymbol{y}_{ij}^t|\boldsymbol{z}_u^t)] - C^t\}\Big]\Big\} - T\log s
\end{aligned}
$$

where $C^t \in \mathbb{R}$ is the maximum value of $\sum_{(i,j)\in\mathbb{Y}} \log[P(\boldsymbol{y}_{ij}^t|\boldsymbol{z}_u^t)]$ over $m$ samples but within a time point $t$.

We also update the penalty parameter $\kappa$ to improve convergence and to reduce reliance on its initialization. In particular, after the $a$-th ADMM iteration, we calculate the respective primal and dual residuals:

$$r_{\text{primal}}^{(a)} = \sqrt{\frac{1}{T\times d}\sum_{t=1}^{T}\|\boldsymbol{\mu}_{(a)}^t - \boldsymbol{\nu}_{(a)}^t\|_2^2} \ \text{ and } \ r_{\text{dual}}^{(a)} = \sqrt{\frac{1}{T\times d}\sum_{t=1}^{T}\|\boldsymbol{\nu}_{(a)}^t - \boldsymbol{\nu}_{(a-1)}^t\|_2^2}.$$

Throughout, we initialize the penalty parameter $\kappa = 10$. We jointly update the penalty parameter $\kappa$ and the scaled dual variable $\boldsymbol{w}$ as in Boyd et al. (2011) with the following conditions:

$$
\begin{aligned}
\kappa_{(a+1)} &= 2\kappa_{(a)}, \quad \boldsymbol{w}_{(a+1)} = \frac{1}{2}\boldsymbol{w}_{(a)}, \quad \text{if } r_{\text{primal}}^{(a)} > 10 \times r_{\text{dual}}^{(a)}, \\
\kappa_{(a+1)} &= \frac{1}{2}\kappa_{(a)}, \quad \boldsymbol{w}_{(a+1)} = 2\boldsymbol{w}_{(a)}, \quad \text{if } r_{\text{dual}}^{(a)} > 10 \times r_{\text{primal}}^{(a)}.
\end{aligned}
$$

### 7.5.2 Post-Processing

Since neural networks may be over-fitted for a statistical model in change point detection, we track the following Coefficient of Variation as a signal-to-noise ratio when we learn the model parameter with the full time series data:

$$\text{Coefficient of Variation} = \frac{\text{mean}(\Delta\hat{\boldsymbol{\mu}})}{\text{sd}(\Delta\hat{\boldsymbol{\mu}})}.$$

We choose the learned parameter $\hat{\boldsymbol{\mu}}$ with the largest Coefficient of Variation as final output.

By convention, we also implement two post-processing steps to finalize the detected change points. When the gap between two consecutive change points is small or $\hat{C}_k - \hat{C}_{k-1} < \epsilon_{\text{spc}}$, we preserve the detected change point with greater $\Delta\hat{\boldsymbol{\zeta}}$ value to prevent clusters of nearby change points. Moreover, as the endpoints of a time span are usually not of interest, we remove the $\hat{C}_k$ smaller than a threshold $\epsilon_{\text{end}}$ and the $\hat{C}_k$ greater than $T - \epsilon_{\text{end}}$. In Section 5, we set $\epsilon_{\text{spc}} = 5$ and $\epsilon_{\text{end}} = 5$.

## 7.6 Real Data Experiments

Besides aligning the detected change points with significant real events for interpretation, we can use the Degree-Corrected Stochastic Block Models (DCSBM) (Karrer & Newman, 2011; Zhao et al., 2012) to heuristically compare the change points detected by the proposed and competitor methods. Specifically, we first remove a pre-specified subset of graphs from the time series, and we fit a DCSBM to the remaining graphs within each interval segmented by two consecutive detected change points. For each removed graph, we compute its log-likelihood value under the fitted DCSBM corresponding to its assigned interval. Heuristically, a higher log-likelihood indicates that the removed graph is more structurally stable within its interval, thereby supporting the validity of the detected change points.

To implement this supplementary evaluation procedure, we use the `nett` package (Amini et al., 2013) in R to fit the DCSBM. For two consecutive change points, we consider the time series between them and we exclude the graphs in a pre-specified subset. The remaining graphs in the interval are used to compute the average adjacency matrix and fitted to the DCSBM. We let the number of community be $K = \{2, 3, 4, 5\}$ and we use the lowest BIC score (Wang & Bickel, 2017) to choose the optimal number of community for that interval. Lastly, we calculate the log-likelihood of the removed graphs with the node labels estimated by the `nett` package.

The Degree-Corrected Stochastic Block Model (DCSBM) extends the Stochastic Block Model (SBM) by incorporating degree heterogeneity. The expected adjacency matrix under the DCSBM is given by $\mathbb{E}(A_{ij}|c) = \theta_i \theta_j P_{c_i,c_j}$, where $\theta_i$ is the degree parameter for node $i$, $c_i \in \{1, \ldots, K\}$ is the community assignment of node $i$, and $P_{c_i,c_j} \in [0,1]^{K \times K}$ is the block probability matrix. The `nett` package fits a DCSBM by maximizing the following conditional log pseudo-likelihood (Amini et al., 2013):

$$l(\pi, \Theta; \{\boldsymbol{b}_i\}) = \sum_{i=1}^{n} \log \Big( \sum_{l=1}^{K} \pi_l \prod_{k=1}^{K} \theta_{lk}^{b_{ik}} \Big).$$

With an initialized label vector $e = (e_1, \ldots, e_n)$ for $e_i \in \{1, \ldots, K\}$, the expectation–maximization (EM) algorithm iteratively implements the following updates until convergence:

$$\pi_{il} = \frac{\pi_l \prod_{m=1}^{K} \theta_{lm}^{b_{im}}}{\sum_{k=1}^{k} \pi_k \prod_{m=1}^{K} \theta_{km}^{b_{im}}}, \quad \pi_l = \frac{1}{n} \sum_{i=1}^{n} \pi_{il}, \quad \theta_{lk} = \frac{\sum_i \pi_{il} b_{ik}}{\sum_i \pi_{il} d_i},$$

with node degree $d_1, \ldots, d_n$. The label vector is updated as $e_i = \arg\max_l \pi_{il}$, and the block sums are calculated as $b_{ik} = \sum_j A_{ij} \cdot 1(e_j = k)$ for node $i = 1, \ldots, n$ and block $k = 1, \ldots, K$.

Once the updated node labels $e = (e_1, \ldots, e_n)$ are obtained, the `nett` package in R calculates the block sum $B_{kl} = \sum_{ij} A_{ij} \cdot 1(e_i = k, e_j = l)$, node parameter $\theta_i = d_i / \sum_j B_{e_i, e_j}$, and the log-likelihood of DCSBM:

$$l(\theta; A) = \sum_{ij} A_{ij} \log(\theta_i \theta_j B_{e_i, e_j}) - \frac{1}{2}\Big( \sum_{k,l} B_{kl} + \sum_k B_{kk}\Big( \sum_{i:e_i=k} \theta_i^2 \Big)\Big) + \sum_k n_k \log\Big(\frac{n_k}{n}\Big).$$

Although this is an heuristic evaluation approach, the log-likelihood values can potentially provide information on how well the detected change points have segmented the entire time span into intervals, in which the network patterns are relatively stable.

In Section 5.3, we apply the evaluation procedure to real data. For the MIT cellphone data with $T = 232$, we remove the graphs at time $t$ equals to multiples of $\Delta t = \{15, 20, 25, 30\}$ respectively. For the Enron Email

data with $T = 100$, we remove the graphs at time $t$ equals to multiples of $\Delta t = \{3, 6, 9, 12\}$ respectively. For completeness, Table 8 provides the dates of detected change points from the competitor methods on the respective MIT cellphone data and Enron email data.

Table 8: Dates of detected change points from the competitor methods.

| Data | Method | Dates of Detected Change Points |
|---|---|---|
| MIT cellphone data | CPDnbs | 2004-10-30, 2004-12-17, 2005-01-18, 2005-02-19, 2005-03-11, 2005-03-31 |
| | CPDrdpg | 2004-12-17, 2005-01-06, 2005-03-11 |
| | CPDstergm | 2004-10-13, 2004-10-24, 2004-11-02, 2004-11-16, 2004-12-17, 2005-03-24 |
| | kerSeg | 2004-10-18, 2004-12-17, 2005-01-05, 2005-04-02 |
| | gSeg | 2004-11-01, 2004-12-11, 2004-12-18, 2005-01-10, 2005-03-11 |
| Enron email data | CPDnbs | 2000-09-04, 2001-02-05, 2001-06-11, 2001-09-03, 2001-11-26 |
| | CPDrdpg | 2000-10-09, 2001-01-08, 2001-05-21, 2001-08-13, 2001-09-17, 2001-12-10, 2002-02-04 |
| | CPDstergm | 2001-08-20, 2001-10-15, 2001-12-10, 2002-02-04 |
| | kerSeg | 2000-07-10, 2000-10-23, 2001-05-21, 2002-02-11, 2002-03-25 |
| | gSeg | 2000-07-10, 2000-10-16, 2001-12-31, 2002-02-11, 2002-03-25 |

