# OpenReview forum: "Change Point Detection in Dynamic Graphs with Decoder-only Latent Space Model"
_TMLR — Accepted by TMLR_

### Review · Reviewer_Fgoe · 2025-02-12

**Summary Of Contributions:**

I was one of the reviewers of the previous submission. Since the main contributions remain the same, this section will be very similar to my previous review. The article studies the problem of Change-Point Detection in the context of graphs. That is to say, we observe a set of $T$ graphs (indexed by $t$ which is assumed to correspond to a certain temporal notion) and we would like to detect if and when the generative process underlying each graph changes.

To this end, the authors propose a generative model, where a latent vector $\mathbf{z}\in\mathbb{R}^d$ is assumed to exist, which in turn generates the graph through an inhomogeneous Bernoulli model. In particular, an edge between nodes $i$ and $j$ exists with probability given by $g(\mathbf{U}\mathbf{U}^\top)$ (for the undirected case), where $\mathbf{U}=MLP(\mathbf{z})$ is a multi-layer perceptron (which maps the input d-dimensional vector to a $n\times k$ matrix, being $n$ the number of nodes) and $g(\cdot)$ is a point-to-point sigmoid.

Instead of trying to find (fit) $\mathbf{z}$ for each $t$, the paper studies an approach very similar to a Variational Autoencoder (VAE), where $\mathbf{z}$ is random and a certain prior distribution is assumed (in this case an uncorrelated Gaussian vector). The problem then becomes how to fit the mean of this distribution $\boldsymbol{\mu}$  for each $t$, and a change-point will be flagged if the change between time-steps is high enough. Actually, two methods are proposed, although the second one, a relatively simple heuristic, proves to work best in practice.

I still believe that the main contribution is presenting an objective function specially tailored for CPD, and a method to solve the resulting optimization problem. Instead of using ELBO as in most VAEs, the authors considered the log-likelihood plus a regularization term that penalizes differences between consecutive $\boldsymbol{\mu}$ (instead of the KL-divergence of ELBO). Experimental results including synthetic as well as real-world data confirm the competitiveness of the proposal w.r.t. other methods.

**Audience:**

Yes

**Claims And Evidence:**

No

**Requested Changes:**

Given my comments above, I would request three major changes:
1. Further verify the expressiveness of the proposed model by measuring and comparing other indicators of the graphs. I would be explicit in the manuscript about the limitations of the proposed method in terms of, at least, its impossibility to express non PSD connectivity matrices (and thus negating heterophilic behavior of the nodes).
2. Comparing against Marenco et al. (2022) should be straightforward and I would add it to the comparison. A comparison to Gong et al. (2023) would require an implementation, but the authors may undertake its implementation and publish it (and BTW, thanks for the code).
3. I insist that further discussion about the detected change-points in the real-world dataset is necessary. A simple graph of the number of edges and non-isolated nodes is enough to see that there are effectively change-points at February and March 2002 which for some reason are ignored by the proposed method due to the chosen threshold.

**Strengths And Weaknesses:**

As I mentioned before, I believe that the main contribution and thus biggest strength of the paper is the proposal of the model and the cost in Equation (2), and the method to solve the resulting optimization problem.

However, the manuscript still has several shortcomings, particularly regarding some of the claims and the provided evidence, some of which are also highlighted in the "Changes Since Last Submission".

* I appreciate the effort by the authors of trying to verify the expressiveness of the proposed method by including the degree distribution of generated as well as actual graphs. However, generating a graph with a given degree distribution is fairly straightforward (e.g. a simple configuration model), and the actual difficulty is to reflect structural features of the graph (e.g. clustering coefficient, shortest paths, etc.). The authors cite the *ergm* package (ref. Hunter et al. (2008b)) to justify why they chose the degree distribution, but in the examples Hunter et al. (2008b) shows several other indicators (see figure 5 of Hunter et al. (2008b), where they display the minimum geodesic distance, edge-wise shared partners, triad census and degree distribution). For further examples see for instance Sec. 6.2.4.2 of the classic book "Statistical Analysis of Network Data" by Eric D. Kolaczyk.

* I also appreciate that the authors have added the CPDrdpg method as another baseline. However, the methods of  Larroca et al (2021) and Marenco et al (2022) can be easily added as their code is publicly available (as indicated in those papers, you can obtain the code from https://github.com/git-artes/cpd_rdpg). In particular, I agree that the former is basically CPDrdpg, but the latter is significantly different. The other method asked for by the previous action editor (Gong et al. (2023)) is pertinent, but an implementation is not available as stated by the authors.

* Authors claim that detected change-points in the real-life data is unreasonable and unrealistic for the other methods. The evidence provided to this end is to measure the likelihood of certain graphs (one every 25 for MIT and one every 7 for Enron, which are removed) under an RDPG model between change-points. To this end, they fit the model to the mean of all but the removed graphs between detected change points. Although interesting, I found some important details missing. How many graphs fall on each observed interval? Why not, for instance, consider all graphs and measure this likelihood for every one, by adjusting in all but this graph (*à la* Leave-one-out cross-validation). But maybe more fundamentally: Why choose an RDPG model when the authors are proposing a new model?

---

> ### Author Response · Authors · 2025-03-15
> **Response to Review Comments**
>
> We are extremely grateful for the constructive comments. We changed the title of our manuscript to ``Change Point Detection in Dynamic Graphs with Decoder-only Latent Space Model", as the paper studies the change point detection problem instead of the graph generation problem. The updated text is colored in blue in the manuscript.
>
> \textbf{Q1:} In this work, we do not propose a new generative model to perform the graph generation task. Instead, we use an existing latent space model $p(y|z)$ that is standard and common in the literature to perform the change point detection task. Although the latent space model $p(y|z)$ is often used for graph generation task (validating the fidelity of generated graphs) or graph modeling task (interpreting the coefficients of fitted model), we can also use the latent space model for change point detection task, by using the decoder $p(y|z)$ to bridge the latent variables $z$ and observed graphs $y$. While many latent space models express a graph $y$ through node level representation $z_i \in \mathbb{R}^d$ for node $i = 1, \dots, n$, we use the latent space model to express a graph $y$ through graph level representation $z \in \mathbb{R}^d$. In brief, we use two common models in the literature to specify the decoder $p(y|z)$ and priors $p(z)$ for our change point detection framework.
>
> In Lines 101 to 102 on Page 3, we write:
>
> "To facilitate change point detection in dynamic graphs, we choose to use the graph decoder that is standard
> and common in the literature (Kipf \& Welling, 2016; Hamilton et al., 2017; Pan et al., 2018; Yang et al.,
> 2019; Chen et al., 2020b; Wang et al., 2021b)."
>
> In Lines 107 to 110 on Page 3, we write:
>
> "While this decoder focuses on homophily (the tendency for similar nodes to connect), an extension to consider heterophily (the tendency for dissimilar nodes to connect) as in Luan et al., (2022), Zhu et al., (2023), Di Francesco et al., (2024), and Luan et al., (2024) is allowed for future development."
>
> Moreover, in Section 5.2 on Page 13, we have also included the figures (Figures 8, 9, 10) of edge-wise shared partners distributions in addition to the degree distributions. In summary, the overall trends of the corresponding distributions for the generated graphs align with those of the simulated graphs.
>
> ---
>
> \textbf{Q2:} This paper studies the offline change point detection problem for dynamic binary networks. In contrast, Marenco et al. (2022) studied the online change point detection problem for dynamic weighted graphs. The goal for an online change point detection method is to detect the single change point as early as possible. Hence the evaluation metrics for these two tasks (online v.s. offline) are substantially different. Furthermore, dynamic weighted networks have edge intensities as extra information about the structures, and the underlying distribution for weighted graphs can be different from that of the binary graphs. The comparison between our method and Marenco et al. (2022) can be difficult to evaluate.
>
>
> Furthermore, Larroca et al. (2021) studied the offline change point detection problem for dynamic weighted networks, which differ from our data type of binary networks. Also, when looking into the repository (\url{https://github.com/git-artes/cpd_rdpg}) provided in their paper, the code for Larroca et al. (2021) is not available. Instead, their code implemented the CPDrdpg method (Madrid Padilla et al, 2022) in Python, so we use the original CPDrdpg (an R package) as an additional competitor in the previous revision.
>
> We have also attempted the method from Gong et al., (2023), as their codes are not publicly available and no email response from the authors for code request. However, their notations in Section 4 is unclear. For example, in Equation (14), the matrix A is equal to matrix A itself multiply by another term involving a matrix divided by another matrix in the fraction form.
>
> Finally, we have added another competitor method, namely CPDnbs (Wang et al., 2021). The CPDnbs method assumes the networks are generated from inhomogeneous Bernoulli models and detects change points by combining sample splitting with wild binary segmentation. Tables 1,2, and 3 display the results from the CPDnbs method for the simulation study. Figures 11 and 12 display the results for real data experiments.

---

> ### Author Response · Authors · 2025-03-15
> **Response to Review Comments**
>
> \textbf{Q3:} In practice, we do not know the true underlying distribution for real data, so there is no general procedure to evaluate the detected change point across different methods. Besides validating the detected change points with significant events as in most of the change point detection literature (see references [11], [12], [18], [19], [20], [21], [22]), we attempt a heuristic approach to compare the results across different detection methods as a supplementary evaluation.
>
> In Lines 311 to 317 on Page 14, we write:
>
> "Specifically, we fit Degree Corrected Stochastic Block Models (DCSBM) (Karrer \& Newman, 2011; Zhao et al., 2012) to the networks between consecutive detected change points, and we evaluate the log-likelihood of out-of-sample networks that were excluded during model fitting. We choose DCSBM, a generalization of Stochastic Block Model (SBM), because  SBM is a well known approximation of graphons, which are among the most general network model in the literature (Airoldi et al., 2013; Olhede \& Wolfe, 2014; Gao et al., 2015). Moreover, DCSBM does not favor either the proposed or competitor methods in terms of fitting the model."
>
>
> Moreover, to assess robustness and sensitivity, we let the time gap $\Delta t$ be varied, and the results are shown in Tables 5 and 7. Specifically, we let $\Delta t = \{15,20,25,30\}$ for the MIT data with $T=232$ and we let $\Delta t = \{3,6,9,12\}$ for the Enron data with $T=100$. On average, about two graphs are removed from each segmented interval for evaluation, and the log-likelihood values from the change points detected by our method are higher most of the time. The previously chosen RDPG model tends to favor the CPDrdpg competitor method, when we varied the time gaps for the removed graphs.
>
>
> We have also discussed the potential reason why change-points at February and March 2002 might not exceed the threshold to be declared by our method, along with a direction for future development.
>
>
> In Lines 395 to 400 on Page 20, we write:
>
> "This discrepancy suggests a potential extension to the proposed framework. We could specify the priors for the graph representations as $z^t \sim \mathcal{N}(\mu^t, \Sigma^t)$ and incorporate the covariance $\Sigma^t$ into the regularization term of the objective function to adaptively detect the changes at different scales. Additionally, the ADMM procedure would require an update for $\Sigma^t$, and the quadratic form for the localization method would need to be modified as $(z^{t} - z^{t-1})^\top (\Sigma^t + \Sigma^{t-1})^{-1}(z^{t} - z^{t-1})\sim \chi_d^2$ when $\mu^t - \mu^{t-1} = 0$. We consider this refinement as a direction for future development."

---

> > ### Comment · Reviewer_Fgoe · 2025-03-19
> > **On the likelihood for DCSBM**
> >
> > Thanks for the update on the manuscript. I have two comments, the first one is regarding the likehood under the DCSBM model. This model is not so well-known, so I would including two things. Firstly, some details regarding how the model was fit (at least what software was used). Secondly, at least an expression for the connection probability (or at least some expression that permits the reader that is not familiar with DCSBM to understand how the likehood is computed).

---

> > > ### Comment · Reviewer_Fgoe · 2025-03-19
> > > **On why change-points might not exceed the threshold**
> > >
> > > My second comment regards the justification as to why change-points at February and March 2002 might not exceed the threshold. I do not understand the connection between adding the covariance into the cost function and the ability to detect those changes.

---

> > > > ### Author Response · Authors · 2025-03-23
> > > > **Response to Review Comments**
> > > >
> > > > \textbf{Q2}: We have provide more information about how we think adding the covariance could potentially help to identify the minor changes as future extension. In particular, in Lines 395-406 on Page 20, we write:
> > > >
> > > > "Besides lowering the threshold to include these change points, the discrepancy suggests a potential extension to improve the proposed framework. Perhaps we could specify the priors for the graph-level representations as $z^t \sim \mathcal{N}(\mu^t, \Sigma^t)$ and incorporate both the mean $\mu^t$ and covariance $\Sigma^t$ into the penalty term of the objective function. For example, to adaptively adjust the changes at different scales, the inverse of the covariance $\Sigma^t \in \mathbb{R}^{d \times d}$ may serve as a scaling factor for the difference $\mu^{t+1} - \mu^{t} \in \mathbb{R}^d$, thereby enhancing the magnitudes. By learning the covariances of the graph-level representations and monitoring their consecutive shifts over time, the latent space regularized in this way could potentially capture more subtle variations in the data space. Furthermore, the ADMM procedure would require an update for $\Sigma^t$, and the quadratic form for the localization method would need to be modified as $(z^{t} - z^{t-1})^\top (\Sigma^t + \Sigma^{t-1})^{-1}(z^{t} - z^{t-1})\sim \chi_d^2$ where the inverse of the covariance is also used for standardization. As there are different ways to incorporate the covariances into the regularization term and learning the positive semi-definite covariance via neural network is challenging, we consider this improvement as a potential direction for future development."

---

> > > ### Author Response · Authors · 2025-03-23
> > > **Response to Review Comments**
> > >
> > > Thanks for the comments!
> > >
> > > \textbf{Q1}: We have included more details about DCSBM in Appendix 7.6 on Page 31, regarding how DCSBM is fitted using the library(nett) package [1] in R and the associated expressions.
> > >
> > > [1] Arash A. Amini, Aiyou Chen, Peter J. Bickel, and Elizaveta Levina. Pseudo-likelihood methods for community detection in large sparse networks. The Annals of Statistics, 2013.

---

> ### Author Response · Authors · 2025-03-15
> **Response to Review Comments**
>
> [1] Thomas N Kipf and Max Welling. Variational graph auto-encoders. arXiv preprint, 2016.
>
> [2] Will Hamilton, Zhitao Ying, and Jure Leskovec. Inductive representation learning on large graphs. Advances in neural information processing systems, 2017.
>
> [3] Shirui Pan, Ruiqi Hu, Guodong Long, Jing Jiang, Lina Yao, and Chengqi Zhang. Adversarially regularized graph autoencoder for graph embedding. arXiv preprint, 2018.
>
> [4] Carl Yang, Peiye Zhuang, Wenhan Shi, Alan Luu, and Pan Li. Conditional structure generation through graph variational generative adversarial nets. Advances in neural information processing systems, 2019.
>
> [5] Zhenxing Chen, Bo Liu, Meiqing Wang, Peng Dai, Jun Lv, and Liefeng Bo. Generative adversarial attributed network anomaly detection. In Proceedings of the 29th ACM International Conference on Information \& Knowledge Management, 2020.
>
> [6] Juexin Wang, Anjun Ma, Yuzhou Chang, Jianting Gong, Yuexu Jiang, Ren Qi, Cankun Wang, Hongjun Fu, Qin Ma, and Dong Xu. scgnn is a novel graph neural network framework for single-cell rna-seq analyses. Nature communications, 2021.
>
> [7] Sitao Luan, Chenqing Hua, Qincheng Lu, Jiaqi Zhu, Mingde Zhao, Shuyuan Zhang, Xiao-Wen Chang, and Doina Precup. Revisiting heterophily for graph neural networks. Advances in neural information processing systems, 2022.
>
> [8] Xiaojing Zhu, Cantay Caliskan, Dino P Christenson, Konstantinos Spiliopoulos, Dylan Walker, and Eric D Kolaczyk. Disentangling positive and negative partisanship in social media interactions using a coevolving latent space network with attractors model. Journal of the Royal Statistical Society Series A: Statistics in Society, 2023.
>
> [9] Andrea Giuseppe Di Francesco, Francesco Caso, Maria Sofia Bucarelli, and Fabrizio Silvestri. Link prediction under heterophily: A physics-inspired graph neural network approach. arXiv preprint, 2024.
>
>
>
> [10] Sitao Luan, Chenqing Hua, Qincheng Lu, Liheng Ma, Lirong Wu, Xinyu Wang, Minkai Xu, Xiao-Wen Chang, Doina Precup, Rex Ying, et al. The heterophilic graph learning handbook: Benchmarks, models, theoretical analysis, applications and challenges. arXiv preprint, 2024.
>
> [11] Oscar Hernan Madrid Padilla, Yi Yu, and Carey E Priebe. Change point localization in dependent dynamic nonparametric random dot product graphs. The Journal of Machine Learning Research, 2022.
>
>
> [12] Daren Wang, Yi Yu, and Alessandro Rinaldo. Optimal change point detection and localization in sparse dynamic networks. The Annals of Statistics, 2021.
>
>
> [13] Brian Karrer and Mark EJ Newman. Stochastic blockmodels and community structure in networks. Physical Review E—Statistical, Nonlinear, and Soft Matter Physics, 2011.
>
> [14] Yunpeng Zhao, Elizaveta Levina, and Ji Zhu. Consistency of community detection in networks under degree-corrected stochastic block models. The Annals of Statistics, 2012.
>
> [15] Edo M Airoldi, Thiago B Costa, and Stanley H Chan. Stochastic blockmodel approximation of a graphon: Theory and consistent estimation. Advances in Neural Information Processing Systems, 2013.
>
> [16] Sofia C Olhede and Patrick J Wolfe. Network histograms and universality of blockmodel approximation. Proceedings of the National Academy of Sciences, 2014.
>
> [17] Chao Gao, Yu Lu, and Harrison H. Zhou. Rate-optimal graphon estimation. The Annals of Statistics, 2015.
>
>
> [18] Rebecca Killick, Idris A Eckley, and Jonathan Philip. A wavelet-based approach for detecting changes in second order structure within nonstationary time series, Electronic Journal of Statistics, 2013.
>
>
>
> [19] Robert Lund, Xiaolan L. Wang, Qi Qi Lu, Jaxk Reeves, Colin Gallagher, and Yang Feng. Changepoint detection in periodic and autocorrelated time series, Journal of Climate, 2007.
>
>
>
> [20] Wang Fan, Wanshan Li, Oscar Hernan Madrid Padilla, Yi Yu, and Alessandro Rinaldo. Multilayer random dot product graphs: Estimation and online change point detection, arXiv preprint, 2023.
>
>
>
> [21] Cappello Lorenzo, and Oscar Hernan Madrid Padilla. Bayesian variance change point detection with credible sets, IEEE Transactions on Pattern Analysis and Machine Intelligence, 2025.
>
>
> [22] Chen Hao, and Nancy Zhang. Graph-based change-point detection, The Annals of Statistics, 2015.

---

### Review · Reviewer_6BDy · 2025-03-02

**Summary Of Contributions:**

This paper presents a generative model-based approach for detecting change points in dynamic graphs by leveraging low-dimensional latent representations and a decoder-only architecture. The incorporation of learnable Gaussian priors and Group Fused Lasso (GFL) regularization effectively highlights significant structural changes while smoothing minor fluctuations. The authors employ Alternating Direction Method of Multipliers (ADMM) for optimization and Langevin Dynamics for posterior inference.

**Audience:**

Yes

**Claims And Evidence:**

Yes

**Requested Changes:**

Questions:
1. As mentioned in the weaknesses, the proposed method is designed for change point detection in specific graph models, whereas the competing methods in the simulation study (Section 5) are nonparametric approaches that do not require strong assumptions. Since the proposed method is naturally expected to perform better on data that aligns with its assumptions, a direct comparison with nonparametric methods may not be entirely appropriate. It would be beneficial to compare the proposed method with other approaches that also target similar types of structured data.
2. Given the distributional assumption on the latent variable z, the proposed method seems to focus primarily on detecting mean shifts in the data. Is it possible to extend this approach to detect changes in variance as well? If so, the model selection process based on the chi-square distribution would likely need to be updated accordingly.
3. The real data examples were evaluated by fitting Random Dot Product Graph models and assessing log-likelihood values. However, can it be guaranteed that the given datasets indeed follow such graph structures? If not, how should the results be interpreted?

**Strengths And Weaknesses:**

Strengths:
This paper presents a method for detecting change points in specific graph models. To this end, by leveraging a generative model, the optimization problem, utilizing Group Lasso and ADMM, is well-posed and appropriately formulated.

Weaknesses:
Conversely, the proposed method tends to work well only for specific graph models. For example, as described in the paper, if the assumption that the latent variable z follows a particular Gaussian distribution or that the resulting graphs are formed based on a Bernoulli distribution does not hold, the performance of the method may deteriorate.

---

> ### Author Response · Authors · 2025-03-15
> **Response to Review Comments**
>
> We are extremely grateful for the constructive comments, particularly for the distributional assumption on the latent variables, which leads to a potential future development. The updated text is colored in blue in the manuscript.
>
> ---
>
> \textbf{Q1:} For the competitor methods, the CPDstergm assumes the graphs are generated from Separable Temporal Exponential Random Graph Model (STERGM), where $y \sim STERGM(\theta)$. The networks in Scenario 1 are directly simulated from STERGM, so Scenario 1 is in favor of CPDstergm. The CPDrdpg assumes the graphs are generated from Random Dot Product Graph (RDPG) model, where $y_{ij} \sim Ber(\theta_{ij} = X_i^\top X_j)$. This model is highly related to Scenario 2, in which the networks are simulated from Stochastic Block Model (SBM). Furthermore, we have added the CPDnbs method (Wang et al., 2021), which assumes the networks are generated from inhomogeneous Bernoulli models with $y_{ij} \sim Ber(\theta_{ij})$. These three competitor methods, along with the proposed CPDlatent method, are targeting different types of structured data.
>
>
>
> The gSeg and kerSeg methods are indeed nonparametric methods, which do not require the graphs to follow a particular graph model. Hence, besides using networks (net.) as input data to their methods, we also use network statistics (stats.) as input for the detection. Though our proposed method has a distributional assumption on the decoder $p(y|z)$ for the graphs, we detect change points based on the graph level representation in the latent space for $p(z)$, which is similar to using network statistics (stats.) as input to gSeg and kerSeg methods.
>
> ---
>
> \textbf{Q2:} Network patterns can be complex due to dyadic dependency, so detecting change points in the data space or $y^t \in \{0,1\}^{n \times n}$ can be challenging. Hence, we turn to detect change points in the latent space or $z^t \in \mathbb{R}^d$, which could potentially be easier with careful design. Essentially, the proposed framework bridges the graphs and representations via a decoder and Group Fused Lasso regularization. Although the changes in network patterns can be different types, when we transfer the information from data space ($y^1, \dots, y^t$) to latent space ($z^1, \dots, z^t$), we want the change signals in networks to be reflected by the mean shifts in the low-dimensional latent space. In other words, we convert the complicated changes in network patterns into simple changes on $\mu^t - \mu^{t-1} \in \mathbb{R}^d$ that are enforced by the Group Fused Lasso penalty.
>
>
> Thanks to this comment, we also have a future direction for this work. In Lines 395 to 400 on Page 20, we write:
>
> "This discrepancy suggests a potential extension to the proposed framework. We could specify the priors for the graph representations as $z^t \sim \mathcal{N}(\mu^t, \Sigma^t)$ and incorporate the covariance $\Sigma^t$ into the regularization term of the objective function to adaptively detect the changes at different scales. Additionally, the ADMM procedure would require an update for $\Sigma^t$, and the quadratic form for the localization method would need to be modified as $(z^{t} - z^{t-1})^\top (\Sigma^t + \Sigma^{t-1})^{-1}(z^{t} - z^{t-1})\sim \chi_d^2$ when $\mu^t - \mu^{t-1} = 0$. We consider this refinement as a direction for future development."

---

> ### Author Response · Authors · 2025-03-15
> **Response to Review Comments**
>
> \textbf{Q3:} In practice, we do not know the true underlying distribution for real data, so we cannot guarantee the given datasets indeed follow such graph structures. Also, the number and location of change points for real data are usually unknown, so there is no widely accepted ground truth or general procedure to evaluate the detected change points across different methods. As a result, besides validating the detected change points with significant events as in most of the literature, we attempt a heuristic approach to compare the results across different detection methods as a supplementary evaluation.
>
>
> In Lines 311 to 317 on Page 14, we write:
>
> "Specifically, we fit Degree Corrected Stochastic Block Models (DCSBM) (Karrer \& Newman, 2011; Zhao et al., 2012) to the networks between consecutive detected change points, and we evaluate the log-likelihood of out-of-sample networks that were excluded during model fitting. We choose DCSBM, a generalization of Stochastic Block Model (SBM), because  SBM is a well known approximation of graphons, which are among the most general network model in the literature (Airoldi et al., 2013; Olhede \& Wolfe, 2014; Gao et al., 2015). Moreover, DCSBM does not favor either the proposed or competitor methods in terms of fitting the model."
>
>
>
> For the two real data experiments, the log-likelihood of out-of-sample graphs using the change points detected from our method are higher most of the time. In other words, the change points from our method segment the time series in a way that better capture the unchanged patterns within each interval. The corresponding results are displayed in Tables 5 and 7. The previously chosen RDPG model tends to favor the CPDrdpg competitor method, when we varied the time gaps for the removed graphs.
>
>
>
> [1] Daren Wang, Yi Yu, and Alessandro Rinaldo. Optimal change point detection and localization in sparse dynamic networks. The Annals of Statistics, 2021.
>
> [2] Brian Karrer and Mark EJ Newman. Stochastic blockmodels and community structure in networks. Physical Review E—Statistical, Nonlinear, and Soft Matter Physics, 2011.
>
> [3] Yunpeng Zhao, Elizaveta Levina, and Ji Zhu. Consistency of community detection in networks under degree-corrected stochastic block models. The Annals of Statistics, 2012.
>
> [4] Edo M Airoldi, Thiago B Costa, and Stanley H Chan. Stochastic blockmodel approximation of a graphon: Theory and consistent estimation. Advances in Neural Information Processing Systems, 2013.
>
> [5] Sofia C Olhede and Patrick J Wolfe. Network histograms and universality of blockmodel approximation. Proceedings of the National Academy of Sciences, 2014.
>
> [6] Chao Gao, Yu Lu, and Harrison H. Zhou. Rate-optimal graphon estimation. The Annals of Statistics, 2015.

---

### Review · Reviewer_kLsW · 2025-03-03

**Summary Of Contributions:**

This paper proposes a generative model that learns a prior distribution for the low-dimensional representation of graph data. This model is then used to detect change points in time series of graphs by analyzing changes in the latent representations.

**Audience:**

Yes

**Claims And Evidence:**

Yes

**Requested Changes:**

Perhaps I overlooked it, but could you please include the environment used to run the method, as well as the time it took for each real data set and simulation?

**Strengths And Weaknesses:**

Compared to the previous version, the new draft addresses most of the previous concerns. The algorithm's complexity has been added, and a new simulation study with proper evaluations has been included, which significantly alleviates previous doubts about the method's power and validity.

---

> ### Author Response · Authors · 2025-03-15
> **Response to Review Comments**
>
> We are extremely grateful for the acknowledgment on the updated manuscript! We have added the environment and running time to the manuscript.
>
> \textbf{Q1:}
>
> In Lines 668-671, we write:
>
> "We run our experiment with Tesla T4 GPU. The running time for the simulated study is about two hours for a scenario with all sequences and cross-validation on the tuning parameter $\lambda$. The running time for the real data experiment is approximately twenty to thirty minutes including cross-validation on the tuning parameter $\lambda$."

---

### Decision · Action_Editor_dkCP · 2025-04-09

**Recommendation:** Accept as is

**Comment:**

This paper is a resubmission of a [previous version](https://openreview.net/forum?id=FVygyCbzon). The reviewers found that this version has been significantly improved and it addresses the previous raised concerns. All three reviewers support acceptance of the paper, and agree that any remaining concerns can be addressed in follow-up research.

The reviewers recognize the promisingness of the method. In particular, they find the model itself, as well as the proposed optimization algorithm, to be the main strength of the paper. Quoting Reviewer Fgoe, the paper introduces "interesting features that open some research avenues".

One concern, raised by Reviewer Fgoe, is on the robustness of the algorithm. In particular, the proposed approach is unable to detect some simple changes in the Enron dataset. The authors have suggested plausible explanations for this, but further evaluation would be needed to confirm these explanations.

**Audience:**

The paper is relevant for researchers interested in change point detection, generative modeling, Group Fused Lasso regularization, etc.

**Claims And Evidence:**

In this paper, the authors put forward an approach, based on generative models, for detecting change points in dynamic graphs. The method makes use of low-dimensional latent variables, a decoder-based architecture, learnable Gaussian priors, and Group Fused Lasso regularization. The approach is demonstrated on simulation studies, which show good performance in change point detection, and on real-data experiements, which shows change points that align with significant events.

Overall, the reviewers found the approach is technically sound and the claims shown in the paper are convicing. The limitations of the approach (see below) have been acknowledged by the authors and can be investigated in future work. All three reviewers found that their concerns have been adequately addressed and support acceptance of the paper.